# Electrocorticographic evidence of a common neurocognitive sequence for mentalizing about the self and others

Kevin M. Tan [ID] [1][✉], Amy L. Daitch[2], Pedro Pinheiro-Chagas[2], Kieran C. R. Fox [ID] [2,3], Josef Parvizi [ID] [2,3] & Matthew D. Lieberman [ID] [1]

Neuroimaging studies of mentalizing (i.e., theory of mind) consistently implicate the default mode network (DMN). Nevertheless, the social cognitive functions of individual DMN regions remain unclear, perhaps due to limited spatiotemporal resolution in neuroimaging. Here we use electrocorticography (ECoG) to directly record neuronal population activity while 16 human participants judge the psychological traits of themselves and others. Self- and other-mentalizing recruit near-identical cortical sites in a common spatiotemporal sequence. Activations begin in the visual cortex, followed by temporoparietal DMN regions, then finally in medial prefrontal regions. Moreover, regions with later activations exhibit stronger functional specificity for mentalizing, stronger associations with behavioral responses, and stronger self/other differentiation. Specifically, other-mentalizing evokes slower and longer activations than self-mentalizing across successive DMN regions, implying lengthier processing at higher levels of representation. Our results suggest a common neurocognitive pathway for self- and other-mentalizing that follows a complex spatiotemporal gradient of functional specialization across DMN and beyond.

[1] Social Cognitive Neuroscience Laboratory, Department of Psychology, University of California, Los Angeles, CA, USA. [2] Laboratory of Behavioral and Cognitive Neuroscience, Department of Neurology and Neurological Sciences, Stanford University, Stanford, CA, USA. [3] School of Medicine, Stanford University, Stanford, CA, USA. ✉email: kevmtan@ucla.edu

Humans are social by nature: our central nervous systems have evolved many mechanisms to support our rich and complex social worlds[1]. Although high levels of sociality are seen throughout the animal kingdom[2,3], humans are exceptional in their capacity for mentalizing: the ability to consider the mental states and traits of others and oneself[4,5]. The field of social neuroscience seeks to understand how mentalizing and other social functions are implemented at the level of brain and biology[6]. In humans, social neuroscience primarily relies on functional magnetic resonance imaging (fMRI), a neuroimaging modality with high spatial resolution but low temporal resolution[7]. Hundreds of fMRI studies have shown that mentalizing recruits default mode network (DMN) regions – including temporoparietal junction (TPJ), posteromedial cortex (PMC), and medial prefrontal cortex (mPFC) – with remarkable consistency across countless mentalizing paradigms instantiated in various sensory modalities[4,8–12]. Nevertheless, the specific social cognitive functions of individual DMN regions remain unclear. When seen through fMRI, DMN regions appear to respond concurrently, yet electrophysiological studies demonstrate that critical neurocognitive dynamics occur at millisecond timescales throughout DMN[13]. Thus, the limited temporal resolution of fMRI may preclude more precise neurocognitive accounts of mentalizing and its component processes.

Several studies have investigated the fast spatiotemporal dynamics of mentalizing processing using source-space electroencephalography (EEG) and magnetoencephalography (MEG), neuroimaging modalities with millisecond temporal resolution but coarse spatial resolution[14]. These studies reveal a general spatiotemporal sequence of cortical recruitment, starting in visual cortex, followed by mirror neuron system regions (MNS; e.g., intraparietal sulcus and premotor cortex), then lastly in DMN regions[15–20]. These findings exemplify the emerging consensus that visual representations are used by MNS to identify observable actions (e.g., grasping for food), which are then used by DMN to infer unobservable mental states (e.g., hunger)[8,21–24]. Taken together, EEG/MEG studies of mentalizing suggest that visual cortex, MNS, and DMN act as a hierarchical neurocognitive pathway that transforms low-level visual inputs into high-level mentalistic inferences. However, despite broad agreement at the network level, these studies report inconsistent recruitment across individual DMN regions. These inconsistencies may reflect limitations in EEG/MEG source localization, particularly in deeper regions such as mPFC and PMC[25], which were not sampled in many of these studies. As such, the sequence of mentalizing processing across individual DMN regions remains unclear.

We sought a more spatiotemporally precise and mechanistic account of mentalizing by exploring neuronal population activity across individual DMN regions and beyond. Leveraging the benefits of human intracranial electrophysiology[26], we recorded high-frequency broadband activity (HFB; 70–180 Hz), which reflects the rapid aggregate spiking of neuronal populations[27]. In contrast, fMRI measures slow metabolic changes, although fMRI and HFB correspond in the anatomy and direction of measured effects (see Parvizi & Kastner, 2019[26]). We show that self- and other-mentalizing share a complex spatiotemporal gradient of functional specialization at millisecond, millimeter, and cross-regional scales. Our findings demonstrate that high spatiotemporal resolution methods can provide critical insights on the neurocognitive mechanisms of human social cognition.

## Results

### Data and design
We recruited sixteen human participants who had electrocorticography (ECoG) electrodes surgically implanted onto the cortical surface for epilepsy monitoring and treatment (Supplementary Table 2). Recordings were obtained from all 2125 electrode sites in our cohort (Fig. 1b). Our behavioral task (Fig. 1a) consisted of true/false text prompts for three conditions of interest: self-mentalizing (e.g., "I am honest"), other-mentalizing (e.g., "My neighbor is honest"), and a non-social 'cognitive' task involving simple arithmetic (e.g., "9 + 86 = 95"; Supplementary Table 4). Sites and trials underwent exclusion criteria for epileptic activity, noise, poor behavioral performance (trials only), and statistical outliers (see Methods).

We began by parcellating the brain into seven regions-of-interests (ROIs; Fig. 1c) using each participant's native-space cortical surface (Supplementary Fig. 2). We included six DMN ROIs that are strongly implicated in mentalizing[12]: temporoparietal junction (TPJ), anterior temporal lobe (ATL), posteromedial cortex (PMC), anteromedial prefrontal cortex (amPFC), dorsomedial prefrontal cortex (dmPFC), and ventromedial prefrontal cortex (vmPFC). Visual cortex was included as a control ROI. Of 2125 electrode sites, 555 were included in our ROIs.

We examined HFB activity from each site in two ways: single-trial and trial-averaged analyzes (Fig. 1d–f). Single-trial analysis compared trialwise HFB responses relative to the pre-stimulus baseline preceding each trial ($p_{FDR} < 0.05$, corrected across timepoints, trials, and sites). Single-trial analysis captured five key metrics of HFB activations: onset, peak, and offset latencies, duration, and peak power (defined in Fig. 1e). Trial-averaged analysis used linear mixed-effects models (LMEMs) to estimate mean timecourses of task-evoked HFB power relative to pre-stimulus baseline (Fig. 1f). Trial-averaged analysis identified sites with significant activations or deactivations for each task condition ($p_{FDR} < 0.05$, corrected across timepoints and sites). See Fig. 2 for within-site analyses of exemplar ROI sites. All statistical tests herein were two-tailed.

### Functional specificity for mentalizing strengthens from visual cortex to mPFC
To explore the fast spatiotemporal dynamics of mentalizing processing, we first examined its functional-anatomic foundations. We began by identifying sites with significant HFB responses to either self- or other-mentalizing, regardless of functional specificity or selectivity (Fig. 3a, b). Using trial-averaged results, sites were considered 'mentalizing-active' (cyan) or 'mentalizing-deactive' (orange) if they produced higher or lower HFB power, respectively, relative to pre-stimulus baseline ($p_{FDR} < 0.05$). Sites were considered 'mentalizing-nonresponsive' if they produced nonsignificant HFB responses to mentalizing. We found mentalizing-active sites in nearly all parts of cortex, while mentalizing-deactive sites were rarer and generally located outside DMN (Fig. 3a). Overall, most cortical sites were mentalizing-nonresponsive (55% whole-brain; Fig. 3b).

Next, we examined the functional specificity of mentalizing-active sites (Fig. 3c, d). Using trial-averaged results, we identified which mentalizing-active sites also produced significant HFB responses to arithmetic (cognitive task) relative to pre-stimulus baseline ($p_{FDR} < 0.05$). Additionally, we directly compared single-trial peak power across mentalizing and arithmetic using robust regression ($p_{FDR} < 0.05$, corrected across sites). Sites were considered 'mentalizing-specific' (light and dark turquoise) if they were (1) mentalizing-active but not arithmetic-active, and (2) produced significantly higher peak power for mentalizing over arithmetic. Sites were considered 'non-specific' (lime, pink, and red) if they coactivated for mentalizing and arithmetic, regardless of peak power differences.

Overall, most mentalizing-active sites were non-specific (68% whole-brain), while the remaining mentalizing-specific sites were unevenly distributed across cortex (Fig. 3c, d). Within our ROIs, the lowest mentalizing-specificity was found in visual cortex (1%).

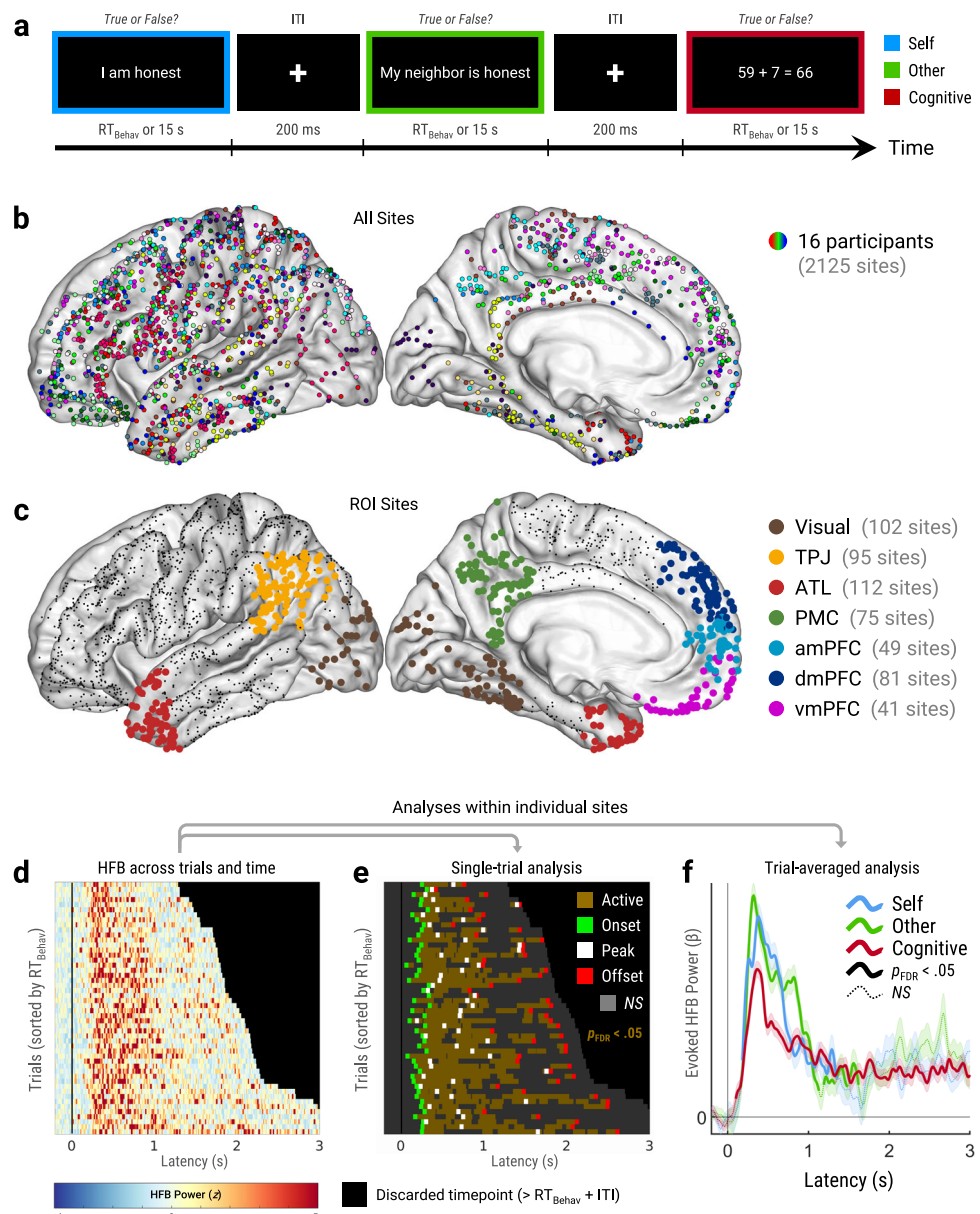

**Fig. 1 Data and design.** Brain maps in main figures plot all sites in left-hemisphere Montreal Neurological Institute space for display purposes. **a** Behavioral task (see Supplementary Table 4). **b** Map of electrode sites colored by participant. **c** Region-of-interest sites (ROI; colored circles) and non-ROI sites (black dots). ROIs were defined using each participant's native cortical surface. **d**–**f** Within-site analytic pipeline using a mid-cingulate exemplar site. **d** High-frequency broadband power (HFB; 70–180 Hz) across timepoints and trials. Black areas indicate discarded timepoints (post-trial). **e** Single-trial analysis compared evoked HFB power versus pre-stimulus baseline (Welch's tests) to provide five key HFB metrics. Duration: total timepoints with significant activations (brown areas; $p_{FDR} < 0.05$, corrected across timepoints, trials, and sites). Onset Latency: earliest timepoint with significant activation (green squares). Peak Latency and Peak Power: timepoint and magnitude, respectively, of the strongest activation (white squares). Offset Latency: latest timepoint with significant activation (red squares). Gray areas indicate nonsignificant (NS) activations. **f** Trial-averaged analysis identified sites that were active, deactive, or nonresponsive for each task condition. Sites were considered 'active' or 'deactive' if evoked HFB power was significantly higher or lower, respectively, than pre-stimulus baseline ($p_{FDR} < 0.05$, corrected across timepoints and sites). Colored lines show mean timecourses of evoked HFB power ($\beta$) estimated by linear mixed-effects modeling (nested within Trial). Thick solid lines indicate significant responses. Thin dashed lines indicate nonsignificant responses. Shaded areas indicate standard error of $\beta$. Abbreviations: s = seconds, ms = milliseconds, ITI = inter-trial interval, $RT_{Behav}$ = behavioral response time, Visual = visual cortex, ATL = anterior temporal lobe, TPJ = temporoparietal junction, PMC = posteromedial cortex, amPFC = anteromedial prefrontal cortex, dmPFC = dorsomedial prefrontal cortex, vmPFC = ventromedial prefrontal cortex, $p_{FDR}$ = p-value adjusted for false discovery rate.

Intermediate mentalizing-specificity was found in TPJ (38%), ATL (32%), and PMC (58%). Very high mentalizing-specificity was found in amPFC (87%), dmPFC (94%), and vmPFC (100%). Taken together, these results show a gradient of mentalizing-specificity from visual cortex to mPFC.

**Mentalizing activations propagate from visual cortex to mPFC.** Next, we explored the timing of mentalizing-evoked activations across ROIs. To this end, we analyzed single-trial HFB latency metrics from mentalizing-active ROI sites (Fig. 3e). Pairwise ROI comparisons (Fig. 3f) measured trial-by-trial ROI differences

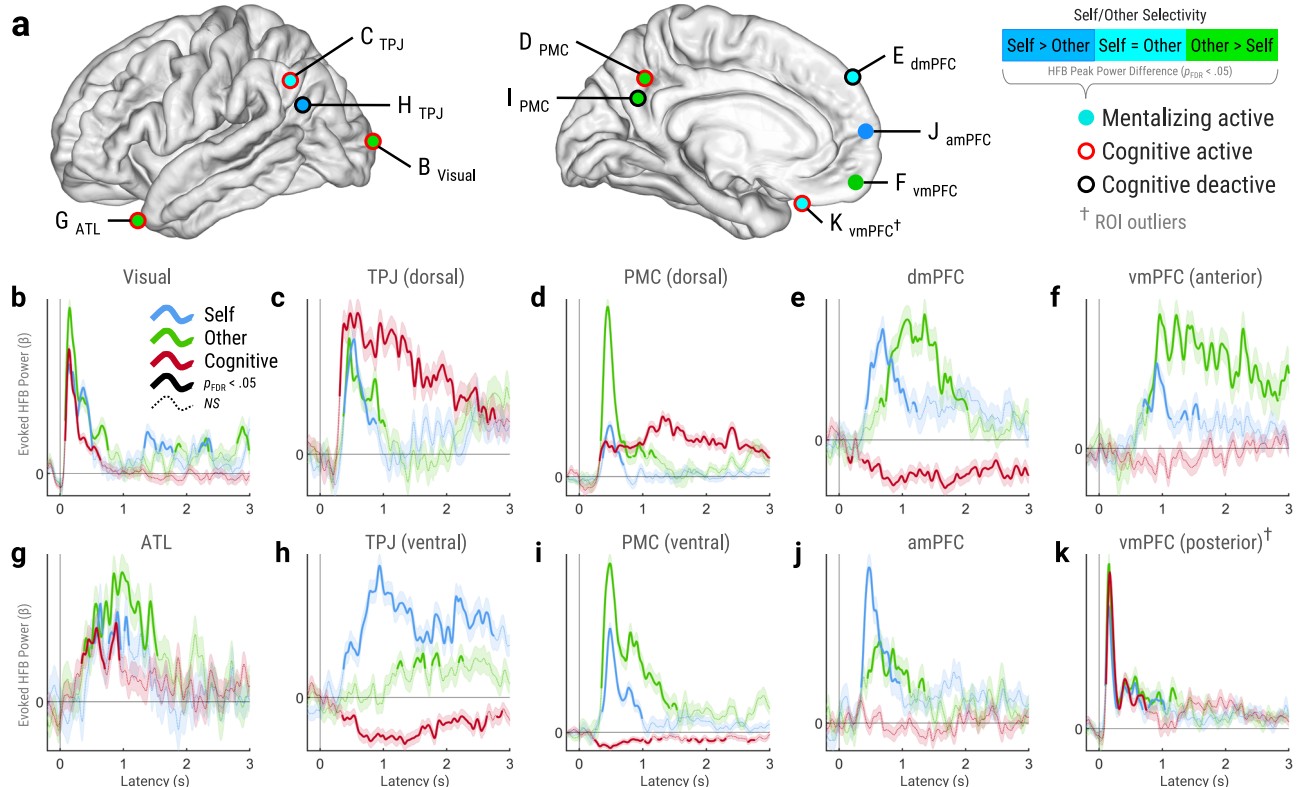

**Fig. 2 Exemplar ROI sites. a** Map of exemplar ROI sites, which were identified as mentalizing-active versus baseline in trial-averaged analysis (see Fig. 1f; $p_{FDR} < .05$, corrected across timepoints and sites). Circle fill color indicates self/other selectivity, which was determined by comparing single-trial HFB peak power (see Fig. 1e) across mentalizing type via robust regression ($p_{FDR} < 0.05$, corrected across sites). Circle outline indicates significant HFB responses to the cognitive (arithmetic) task versus baseline ($p_{FDR} < 0.05$, corrected across timepoints and sites). **b–k** Trial-averaged timecourses of evoked HFB power ($\beta$) from exemplar sites in panel **a**. Thick solid lines indicate significant responses versus baseline ($p_{FDR} < 0.05$, corrected across timepoints and sites). Thin dashed lines indicate nonsignificant (NS) responses. Shaded areas indicate standard error of $\beta$. †Excluded from ROI analyzes due to outlier thresholds (see Methods). *Abbreviations*: HFB = high-frequency broadband (70–180 Hz). ROI = region-of-interest, Visual = visual cortex, ATL = anterior temporal lobe, TPJ = temporoparietal junction, PMC = posteromedial cortex, amPFC = anteromedial prefrontal cortex, dmPFC = dorsomedial prefrontal cortex, vmPFC = ventromedial prefrontal cortex. Source data are provided as a Source Data file.

while controlling for behavioral response times ($RT_{Behav}$) and participant-related heterogeneity using LMEMs (nested within Trial within Participant). Critically, pairwise comparisons only included participants with mentalizing-active sites in both ROIs.

We found that visual cortex produced the earliest activation onsets ($M = 101 \pm 1$ ms) of any ROI (Fig. 3f; $p_{FDR} < 0.05$, corrected across unique ROI pairs). Afterwards, mid-latency onsets were produced by TPJ ($M = 303 \pm 7$ ms), ATL ($M = 316 \pm 7$ ms), and PMC ($M = 322 \pm 5$ ms), with nonsignificant differences between them. Later onsets were produced by amPFC ($M = 465 \pm 10$ ms) and dmPFC ($M = 466 \pm 7$ ms), with nonsignificant differences between them, followed lastly by vmPFC ($M = 537 \pm 15$ ms). Critically, propagation of activations across ROIs had robust trial-by-trial consistency (Supplementary Fig. 3). Peak latencies showed a similar pattern of cross-ROI differences as onsets, although intra-mPFC differences were nonsignificant (Fig. 3f). Offset latencies showed the least cross-ROI differentiation. Earlier offsets were produced by visual cortex, TPJ, and ATL, with nonsignificant differences between them. Later offsets were produced by amPFC, dmPFC, and vmPFC, with nonsignificant differences between them. We also performed post-hoc analysis of offset latencies relative to $RT_{Behav}$ (Fig. 4c), revealing that mPFC activations more closely preceded $RT_{Behav}$ than other ROIs combined ($b_{20204} = 138 \pm 31$ ms, $p = 4.28e-6$). Nevertheless, despite these robust cross-ROI latency differences, ROIs predominantly activated at overlapping

times (Figs. 2, 3e, 5c and 6; single-participant results in Supplementary Fig. 2).

In sum, mentalizing evoked largely concurrent activations across ROIs, as might be expected from neuroimaging literature. Nonetheless, fine-scale cross-ROI differences in onset, peak, and offset latencies depict an overarching spatiotemporal sequence of activation from visual cortex to mPFC. This sequence extended across individual sites throughout cortex (Fig. 4a).

**A spatiotemporal gradient of mentalizing-specificity from visual cortex to mPFC.** Thus far, we have found spatial gradients in the timing (Fig. 3e, f) and functional specificity (Fig. 3c, d) of neuronal population responses to mentalizing. To examine the correspondence between these gradients, we used logistic mixed-effects classification to predict mentalizing-specificity from mean onset latencies across mentalizing-active ROI sites, nested within Participant (Fig. 4b). We found that the probability of mentalizing-specificity over non-specificity rose by 1.4% per one-millisecond increase in onsets (odds ratio = 1.014, $b_{255} = 0.014 \pm 0.002$, $p = 9.21e-11$). This robust effect was largely attributable to between-ROI differences, as within-ROI effects were weaker and significant only in TPJ ($p = 0.047$), ATL ($p = 0.039$), and PMC ($p = 0.005$). In sum, ROI sites with later activations were more likely to be mentalizing-specific.

These results portray a neurocognitive sequence[28–31] that reflects both the timing and specificity of mentalizing processing

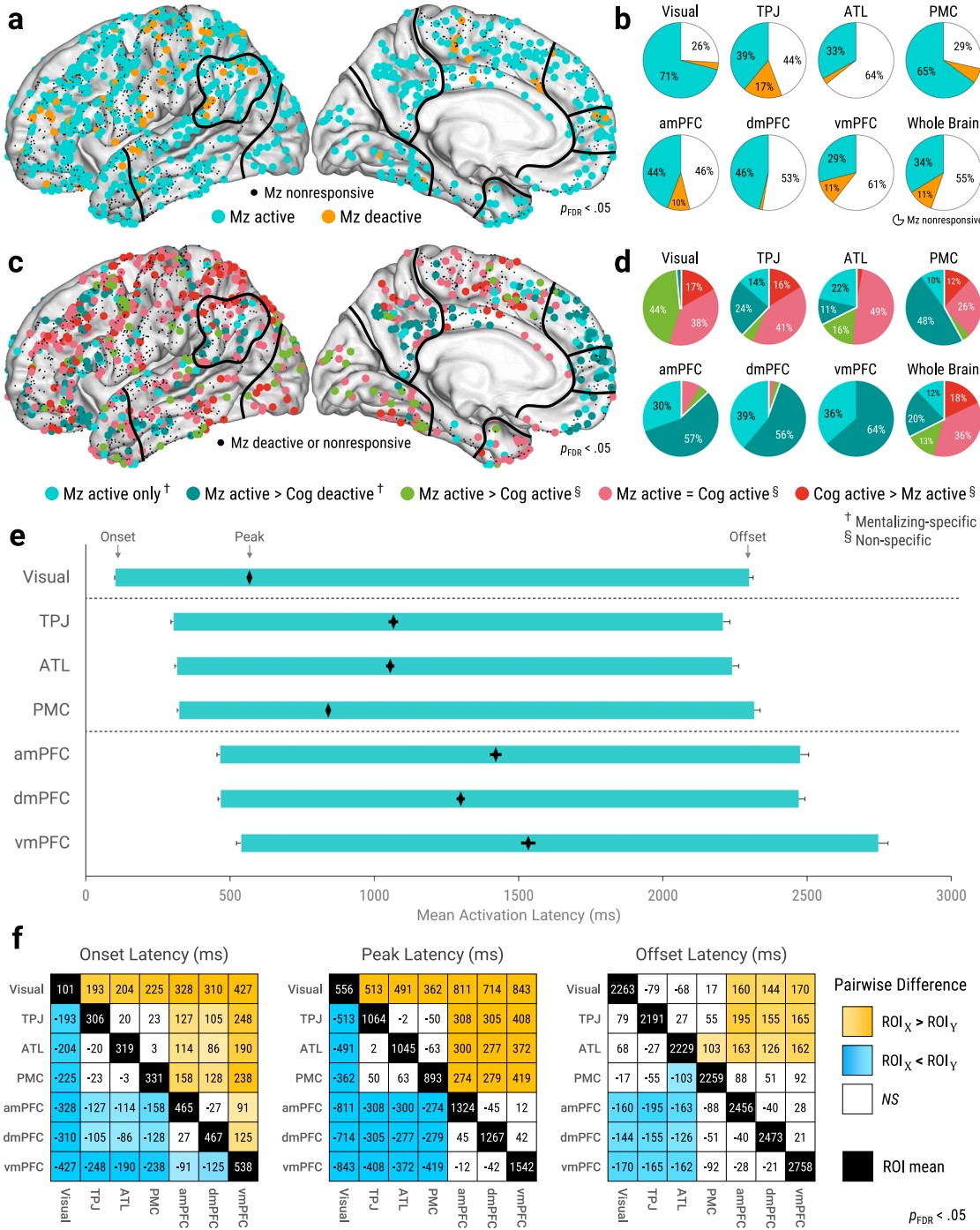

**Fig. 3 A neurocognitive sequence for mentalizing.** Brain maps plot all sites on left hemisphere with approximate ROI outlines (full views in Supplementary Figs. 6 and 7). **a, b** Sites identified as active, deactive, or nonresponsive for mentalizing versus baseline (see Figs. 1f and 2; $p_{FDR} < 0.05$, corrected across timepoints and sites). **c, d** Functional specificity of mentalizing-active sites. [†]Mentalizing-specific sites, defined as mentalizing-active but not cognitive-active ($p_{FDR} < 0.05$, corrected across timepoints and sites) with significantly higher peak power for mentalizing ($p_{FDR} < 0.05$, corrected across sites). [§]Non-specific sites, defined as mentalizing-active and cognitive-active. **e** Mean ROI activation latencies. Left and right floating bar edges depict onsets and offsets, respectively, while diamonds depict peaks (see Fig. 1e). Error bars depict standard error of the mean. **f** Pairwise ROI comparisons of activation latencies. Black diagonal squares show ROI means. Off-diagonal squares show estimates for ROI_X - ROI_Y (controlled for behavioral response times). Orange and blue squares indicate significant differences; color intensity indicates effect size ($p_{FDR} < 0.05$, corrected across unique ROI pairs). White squares indicate nonsignificant (*NS*) differences. Pairwise comparisons used linear mixed-effect models restricted to participants with mentalizing-active sites in both ROIs (nested within Trial within Participant). See Table 1 for *n* of panels **e**, **f**. *Abbreviations*: Mz = mentalizing (collapsed across self and other), Cog = cognitive task (arithmetic), Visual = visual cortex, ATL = anterior temporal lobe, TPJ = temporoparietal junction, PMC = posteromedial cortex, amPFC = anteromedial prefrontal cortex, dmPFC = dorsomedial prefrontal cortex, vmPFC = ventromedial prefrontal cortex, Whole Brain = all relevant sites in cortex, ROI = region-of-interest. Source data are provided as a Source Data file.

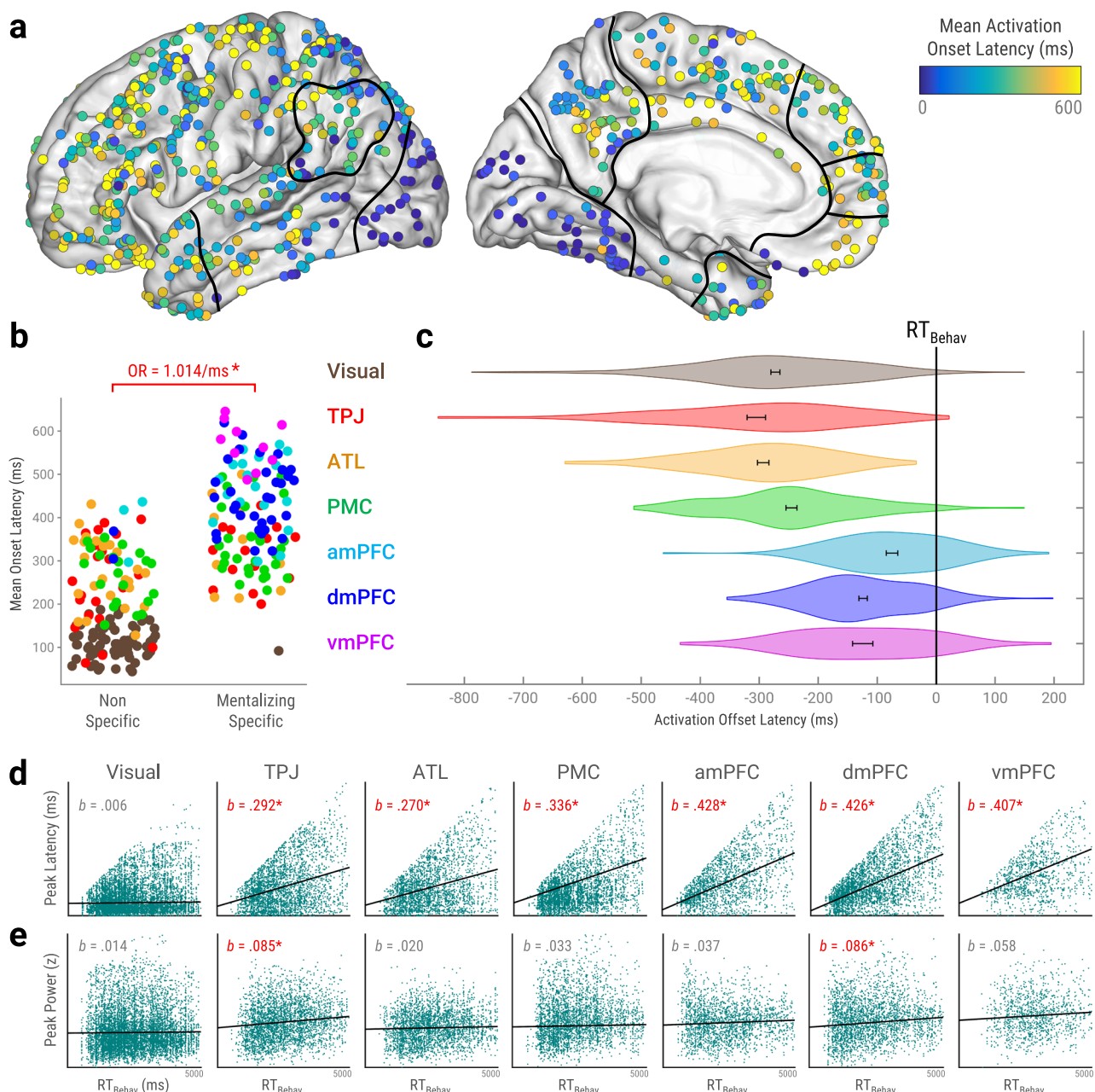

**Fig. 4 Spatiotemporal gradients of functional and behavioral relevance for mentalizing. a** Mean onset latencies of mentalizing-active sites ($p_{FDR} < 0.05$, corrected across timepoints and sites). All sites plotted on left hemisphere with approximate ROI outlines (full views in Supplementary Fig. 8). **b** Mean activation onset latencies of ROI sites by functional specificity (see Fig. 3c). Color indicates ROI. Logistic mixed classification estimated the odds ratio (OR) of mentalizing-specificity over non-specificity per millisecond increase in onsets ($p = 9.21e{-}11$, $n = 257$ sites). **c** $RT_{Behav}$ versus ROI activation offset latencies. Error bar indicates standard error of the mean. Offsets in medial prefrontal cortex (mPFC) ROIs more closely preceded $RT_{Behav}$ than other ROIs combined ($b_{20204} = 138 \pm 31$ ms, $p = 4.28e{-}6$, two-tailed), as per post-hoc LMEM. **d**, **e** Associations between $RT_{Behav}$ and peak activation metrics within ROIs. Slope estimates (*b*) were controlled for behavioral response choices and stimulus visual dissimilarity via LMEMs. See Table 1 for exact statistics and *n*. **d** $RT_{Behav}$ and activation peak latency. Slopes (*b*) estimate change in peak latency per millisecond increase in $RT_{Behav}$. Horizontal slopes (*b* = 0) would indicate purely stimulus-locked activity, while 45° slopes (*b* = 1) would indicate purely $RT_{Behav}$-locked activity (see DiCarlo & Maunsell, 2005[106]). **e** $RT_{Behav}$ and peak power (activation magnitude). Slopes (*b*) estimate change in peak power per 1000 ms increase in $RT_{Behav}$. *Significant OR and *b* estimates have red font (*p* < 0.05, uncorrected, two-tailed). *Abbreviations*: $RT_{Behav}$ = behavioral response time. ROI = region-of-interest, Visual = visual cortex, ATL = anterior temporal lobe, TPJ = temporoparietal junction, PMC = posteromedial cortex, amPFC = anteromedial prefrontal cortex, dmPFC = dorsomedial prefrontal cortex, vmPFC = ventromedial prefrontal cortex, z = z-score, LMEM = linear mixed-effect model. Source data are provided as a Source Data file.

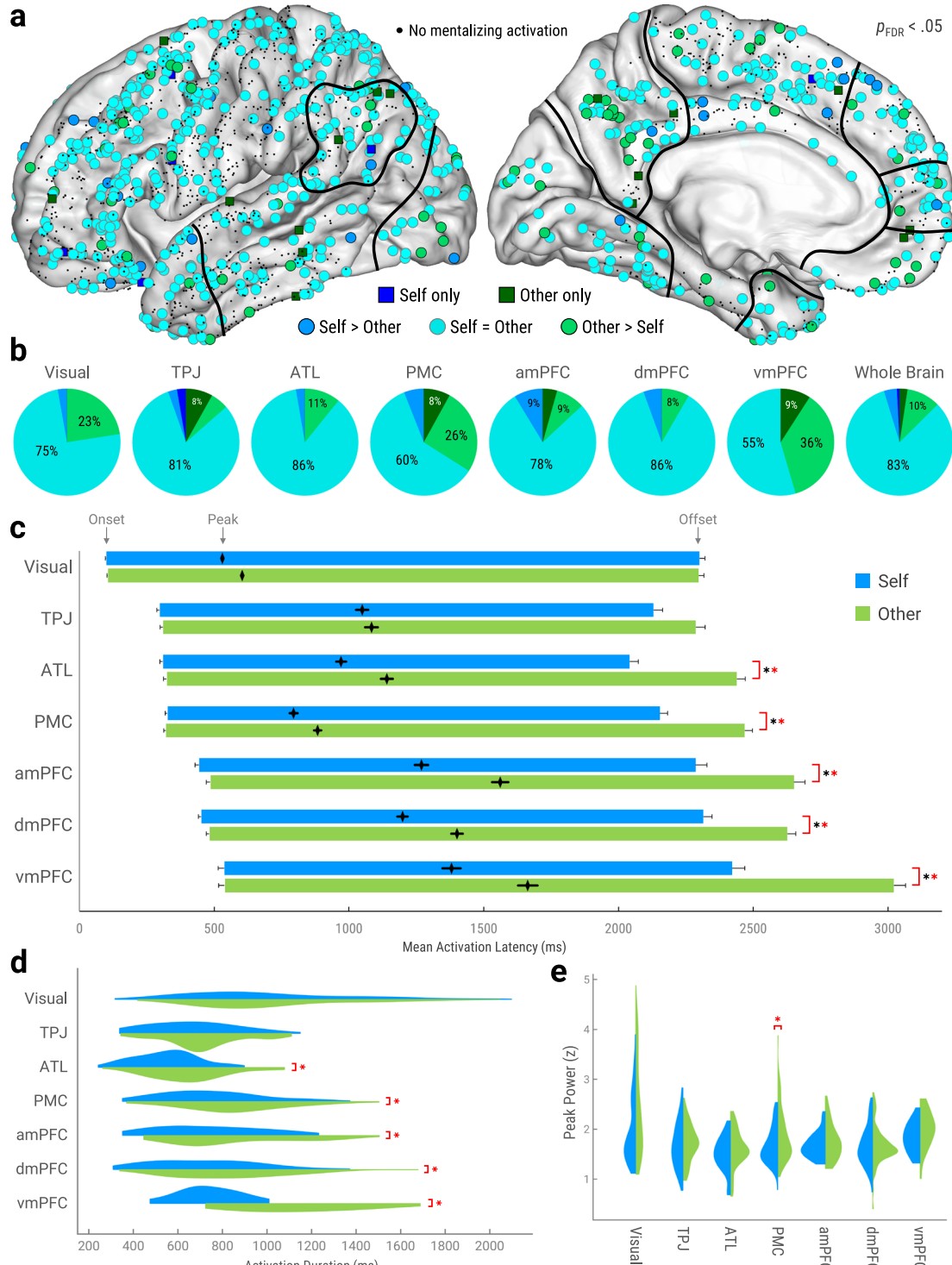

**Fig. 5 A common neurocognitive sequence for self- and other-mentalizing. a** Functional anatomy of self- and other-mentalizing. Circles represent sites identified as coactive for both mentalizing types versus baseline ($p_{FDR} < 0.05$, corrected across timepoints and sites), colored by self/other differences in peak power ($p_{FDR} < 0.05$, corrected across sites; see Fig. 2). Squares represent sites identified as active for only one mentalizing type with significantly higher peak power for that mentalizing type. Black dots represent sites with nonsignificant mentalizing activations. All sites plotted on left hemisphere with approximate ROI outlines (full views in Supplementary Fig. 9). **b** Proportions of mentalizing-active ROI sites exhibiting the selectivity profiles in **a**. 'Whole Brain' refers to all mentalizing-active sites in cortex. **c** Mean activation latencies by mentalizing type and ROI. Left and right floating bar edges depict onsets and offsets, respectively, while diamonds depict peaks (see Fig. 1e). Error bars depict standard error of the mean. *Significant self/other differences for peaks (black) and offsets (red); $p < 0.05$, uncorrected, controlled for $RT_{Behav}$ and stimulus visual dissimilarity, two-tailed. Activation duration (**d**) and magnitude (**e**; peak power) by mentalizing type and ROI. *Red asterisks indicate significant self/other differences ($p < 0.05$, uncorrected, controlled for $RT_{Behav}$ and stimulus visual dissimilarity, two-tailed). See Table 1 for exact statistics and $n$ for **c**–**e**. Abbreviations: ROI = region-of-interest, Visual = visual cortex, ATL = anterior temporal lobe, TPJ = temporoparietal junction, PMC = posteromedial cortex, amPFC = anteromedial prefrontal cortex, dmPFC = dorsomedial prefrontal cortex, vmPFC = ventromedial prefrontal cortex, z = z-score. Source data are provided as a Source Data file.

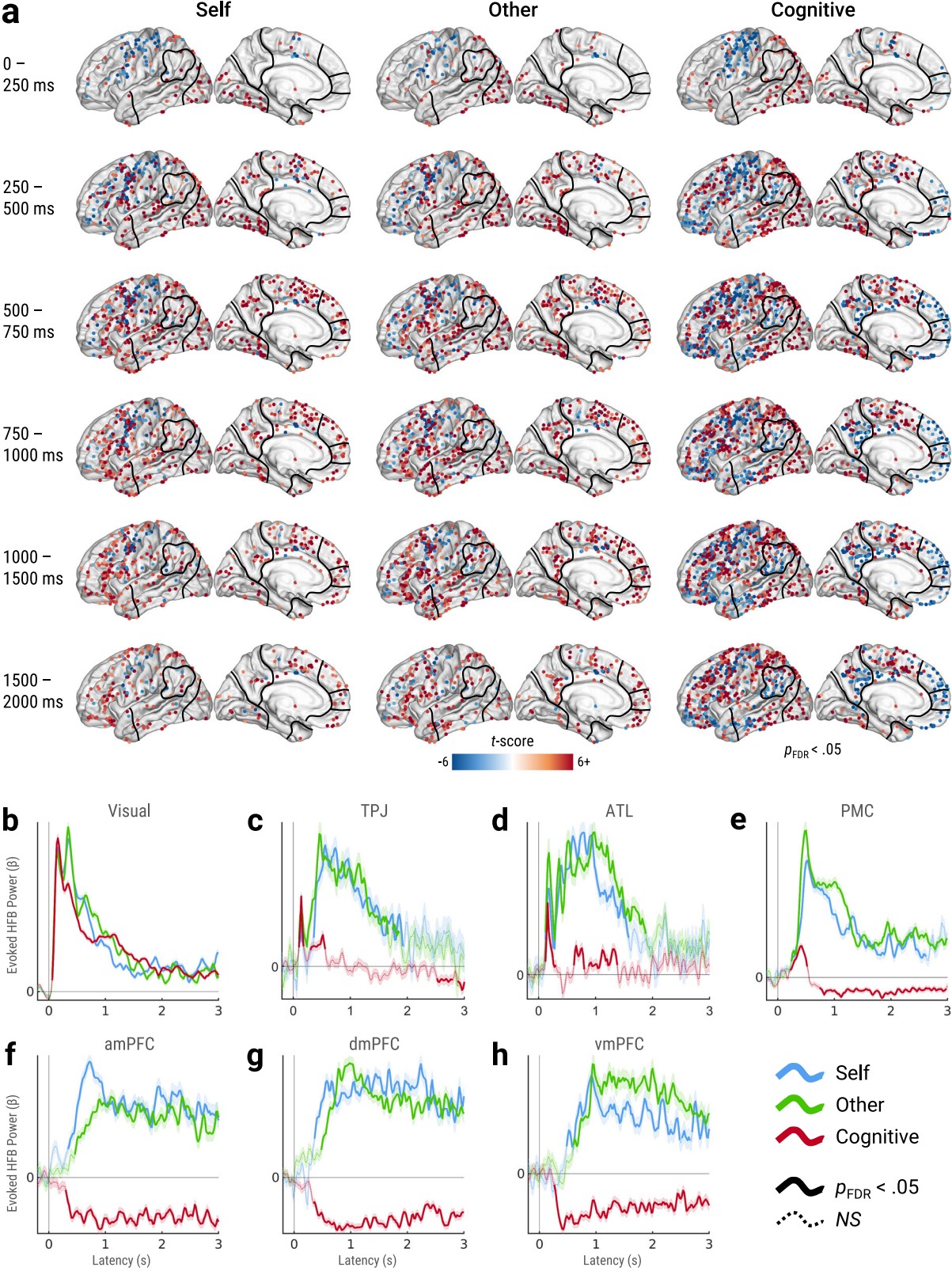

**Fig. 6 Summary of evoked neuronal activity. a** Sites with significant HFB responses during specific time windows versus pre-stimulus baseline, colored by LMEM $t$-values ($p_{FDR} < 0.05$, corrected across sites and time windows, two-tailed). All sites plotted on left hemisphere with approximate ROI outlines. **b–h** Grand-average timecourses of evoked HFB power ($\beta$) within ROIs (LMEM-estimated). Thick solid lines indicate significant responses versus pre-stimulus baseline ($p_{FDR} < 0.05$, corrected across ROIs and timepoints, two-tailed). Thin dashed lines indicate nonsignificant responses. Shaded areas indicate standard error of $\beta$. Abbreviations: HFB = high-frequency broadband (70–180 Hz), LMEM = linear mixed-effects model, ROI = region-of-interest, Visual = visual cortex, ATL = anterior temporal lobe, TPJ = temporoparietal junction, PMC = posteromedial cortex, amPFC = anteromedial prefrontal cortex, dmPFC = dorsomedial prefrontal cortex, vmPFC = ventromedial prefrontal cortex. Source data are provided as a Source Data file.

across ROIs. Within this sequence, visual cortex appears to be a pre-mentalizing stage, as it produced the earliest activations and negligible mentalizing-specificity. Next, temporoparietal DMN regions (TPJ, ATL, and PMC) appear to be lower-level mentalizing stages, as they produced mid-latency activations and intermediate mentalizing-specificity. Finally, mPFC regions (amPFC, dmPFC, and vmPFC) appear to be higher-level mentalizing stages, as they featured the latest activations and overwhelming mentalizing-specificity.

**Behavioral responses are best predicted by TPJ and dmPFC activity.** To explore the relationships between neuronal and behavioral responses, we determined which ROIs predicted $RT_{Behav}$ and $Choice_{Behav}$ (choosing 'true' or 'false' for task prompts) by analyzing single-trial HFB metrics from mentalizing-active ROI sites. We used LMEMs to simultaneously estimate neuronal associations with $RT_{Behav}$ and $Choice_{Behav}$ while controlling for stimulus visual dissimilarity (see Supplementary Methods), nested within Site and Participant. We found that onset latencies had significant ($p < 0.05$) positive associations with $RT_{Behav}$ in visual cortex, TPJ, PMC, amPFC, and dmPFC (Table 1). Moreover, we found significant positive associations between $RT_{Behav}$ and peak latencies (all DMN ROIs; Fig. 4d), offset latencies (all ROIs; Table 1), and activation duration (all ROIs). In contrast, activation magnitude (peak power) had significant $RT_{Behav}$ associations in only two ROIs: TPJ and dmPFC (Fig. 4e). Intriguingly, dmPFC was the only ROI that significantly predicted $Choice_{Behav}$ (Table 1). Taken together, behavioral responses were best predicted by TPJ and dmPFC, which had more numerous significant neurobehavioral associations than other ROIs.

Next, we examined whether behavioral responses were better predicted by activity in successive ROI sites. Specifically, we compared mean onset latencies (Fig. 4a) and within-site $RT_{Behav}$ random effect sizes (b from analyses in previous paragraph) using LMEMs nested within Participant. We found that sites with later onsets had stronger $RT_{Behav}$ effects for peak latency ($b_{255} = 0.972 \pm 0.094$, $p = 1.51e\text{-}23$) but not peak power ($b_{255} = 0.019 \pm 0.074$, $p = 0.796$). In sum, $RT_{Behav}$ was better predicted by successive ROIs in terms of activation latency (Fig. 4d) but not activation magnitude (Fig. 4e).

**Self- and other-mentalizing share a common neuroanatomical basis.** To fractionate mentalizing's neurocognitive sequence across mentalizing type, we first explored the anatomical inter-relations between self- and other-mentalizing (Fig. 5a, b). We therefore identified sites that produced significant trial-averaged activations for each mentalizing type relative to baseline ($p_{FDR} < 0.05$, corrected across timepoints and sites). We also directly compared self/other differences in single-trial peak power using robust regression ($p_{FDR} < 0.05$, corrected across sites). Sites were considered 'self-only' or 'other-only' if they (1) produced significant trial-averaged activations for only one mentalizing type, and (2) produced significantly greater peak power for that mentalizing type over another. Sites that activated for both mentalizing types were labeled by peak power differences: 'self-greater' (Self > Other), 'other-greater' (Other > Self), or 'non-selective' (Self = Other).

We found that mentalizing-active sites were overwhelmingly coactive for both mentalizing types (non-selective + self-greater + other-greater = 97% whole-brain; Fig. 5a,b). Moreover, non-selective sites formed the largest category in all ROIs and the whole-brain (range: 55–86%). We also compared amounts of 'self-selective' sites (self-only + self-greater) versus 'other-selective' sites (other-only + other-greater) using McNemar $\chi^2$ (Yates-

corrected; $df = 1$). This revealed that other-selective sites significantly ($p < 0.05$) outnumbered self-selective sites in visual cortex (3% self/23% other; $\chi^2 = 9.39$), PMC (6% self/34% other; $\chi^2 = 8.45$), and the whole-brain (5% self/13% other; $\chi^2 = 24.40$). Nonsignificant self/other differences were found in TPJ (5% self/14% other; $\chi^2 = 0.57$), ATL (3% self/11% other; $\chi^2 = 0.37$), amPFC (9% self/13% other; $\chi^2 = 0$), dmPFC (6% self/8% other; $\chi^2 = 0$), and vmPFC (0% self/45% other; $\chi^2 = 3.20$).

In sum, self- and other-mentalizing recruited near-identical cortical sites in a largely non-selective manner, though selective sites were mostly other-selective in visual cortex and PMC. Unexpectedly, other-selective and self-selective sites did not significantly outnumber each other in TPJ, amPFC, and dmPFC, which contradicts previous fMRI work from our group[32] and elsewhere[33–35].

**Other-mentalizing evokes slower and lengthier activations across successive ROIs.** Given the highly overlapping neuroanatomy of self- and other-mentalizing, we explored whether self/other differences might be better characterized by fast spatiotemporal functional dynamics. We therefore examined single-trial HFB metrics from mentalizing-active ROI sites. Mentalizing type was analyzed by LMEMs that controlled for $RT_{Behav}$ and visual dissimilarity, nested within Site and Participant.

We found that self- and other-mentalizing evoked a common spatiotemporal sequence of HFB activations across ROIs (Fig. 5c). Within this sequence, onset latencies showed nonsignificant self/other differences (Table 1). However, other-mentalizing evoked significantly ($p < 0.05$) later peaks and offsets than self-mentalizing in all DMN ROIs except TPJ (Fig. 5c). Concordantly, activation duration was significantly longer for other- versus self-mentalizing in all DMN ROIs except TPJ (Fig. 5d). In contrast, self/other differences in activation magnitude were significant only in PMC, which produced greater peak power for other-mentalizing (Fig. 5e). Crucially, although HFB metrics often reflected $RT_{Behav}$ and visual dissimilarity, significant self/other differences were ultimately dissociable from these covariates (Supplementary Table 1).

Next, we examined whether self/other differences became stronger in successive ROI sites. We therefore compared mean onset latencies (Fig. 4a) with single-site random effect sizes (b from analyses in previous paragraph) for peak latency (Fig. 5c) and peak power (e.g., self/other selectivity; Fig. 5e) using LMEMs nested within Participant. We found that later onsets predicted stronger self/other differences in peak latencies ($b_{255} = 0.404 \pm 0.131$, $p = 0.002$). However, onsets did not significantly predict peak power ($b_{255} = -0.303 \pm 0.0744$, $p = 0.796$).

In sum, we found that self- and other-mentalizing recruited near-identical sites in a common spatiotemporal sequence (Fig. 5a–c). Within this sequence, other-mentalizing evoked slower (Fig. 5c) and lengthier (Fig. 5d) activations in all DMN ROIs except TPJ. Intriguingly, successive ROI sites had greater self/other differentiation in the timing, rather than selectivity, of activations. Taken together, self/other functional differences were primarily characterized by the timing of activations throughout successive DMN regions.

**Summary of evoked neuronal activity.** To summarize task-evoked neuronal population activity (Fig. 6a), we identified sites with significant HFB responses during specific time windows relative to pre-stimulus baseline ($p_{FDR} < 0.05$, corrected across sites and time windows). From 0-250 ms, activations were largely localized to visual cortex with negligible task condition differences. From 250–500 ms, activations also encompassed temporoparietal and lateral frontal regions, where mentalizing and

**Table 1 Aggregate ROI analyses of mentalizing.**

| Type | Onset (ms) b | SE | p | Peak (ms) b | SE | p | Offset (ms) b | SE | p | Duration (ms) b | SE | p | Peak Power (z) b | SE | p |
|---|---|---|---|---|---|---|---|---|---|---|---|---|---|---|---|
| **Other-Self** | | | | | | | | | | | | | | | |
| Visual | −4 | 3 | 0.230 | 2 | 7 | 0.731 | 10 | 14 | 0.456 | 7 | 19 | 0.718 | 0.025 | 0.013 | 0.058 |
| TPJ | 1 | 14 | 0.962 | 11 | 46 | 0.808 | 2 | 28 | 0.951 | 4 | 28 | 0.886 | 0.002 | 0.017 | 0.901 |
| ATL | 16 | 18 | 0.376 | 80 | 30 | **0.007** | 58 | 26 | **0.026** | 71 | 28 | **0.013** | 0.021 | 0.012 | 0.091 |
| PMC | −12 | 7 | 0.088 | 71 | 15 | **<0.001** | 112 | 18 | **<0.0001** | 101 | 40 | **0.011** | 0.251 | 0.045 | **<0.001** |
| amPFC | 34 | 28 | 0.230 | 182 | 30 | **<0.001** | 151 | 13 | **<0.001** | 158 | 21 | **<0.001** | 0.013 | 0.074 | 0.857 |
| dmPFC | 20 | 18 | 0.279 | 189 | 45 | **<0.001** | 130 | 9 | **<0.001** | 130 | 20 | **<0.001** | 0.023 | 0.041 | 0.569 |
| vmPFC | 21 | 37 | 0.571 | 203 | 63 | **0.001** | 160 | 18 | **<0.001** | 193 | 52 | **<0.001** | 0.147 | 0.154 | 0.341 |
| **RT Behav** | b | SE | p | b | SE | p | b | SE | p | b | SE | p | b | SE | p |
| Visual | 0.004 | 0.002 | **0.021** | 0.006 | 0.004 | 0.143 | 0.897 | 0.006 | **<0.001** | 0.239 | 0.015 | **<0.001** | 0.014 | 0.008 | 0.106 |
| TPJ | 0.025 | 0.005 | **<0.001** | 0.292 | 0.023 | **<0.001** | 0.904 | 0.014 | **<0.001** | 0.143 | 0.016 | **<0.001** | 0.085 | 0.018 | **<0.001** |
| ATL | 0.009 | 0.008 | 0.272 | 0.270 | 0.023 | **<0.001** | 0.772 | 0.015 | **<0.001** | 0.096 | 0.011 | **<0.001** | 0.020 | 0.012 | 0.086 |
| PMC | 0.017 | 0.007 | **0.012** | 0.336 | 0.493 | **<0.001** | 0.853 | 0.023 | **<0.0001** | 0.209 | 0.012 | **<0.001** | 0.033 | 0.018 | 0.060 |
| amPFC | 0.041 | 0.015 | **0.006** | 0.428 | 0.024 | **<0.001** | 0.970 | 0.008 | **<0.001** | 0.158 | 0.018 | **<0.001** | 0.037 | 0.021 | 0.076 |
| dmPFC | 0.038 | 0.008 | **<0.001** | 0.426 | 0.021 | **<0.001** | 0.987 | 0.007 | **<0.001** | 0.183 | 0.015 | **<0.001** | 0.086 | 0.013 | **<0.001** |
| vmPFC | 0.010 | 0.019 | 0.604 | 0.407 | 0.105 | **<0.001** | 0.987 | 0.015 | **<0.001** | 0.197 | 0.041 | **<0.0001** | 0.058 | 0.090 | 0.521 |
| **Choice True-False** | b | SE | p | b | SE | p | b | SE | p | b | SE | p | b | SE | p |
| Visual | 0 | 3 | 0.991 | −21 | 12 | 0.079 | −11 | 8 | 0.141 | −21 | 14 | 0.138 | −0.025 | 0.013 | 0.058 |
| TPJ | −2 | 14 | 0.911 | −14 | 28 | 0.616 | −20 | 21 | 0.347 | −16 | 25 | 0.509 | 0.021 | 0.031 | 0.504 |
| ATL | −9 | 20 | 0.651 | −21 | 34 | 0.548 | −12 | 21 | 0.563 | −21 | 20 | 0.278 | −0.017 | 0.022 | 0.447 |
| PMC | −10 | 10 | 0.355 | −5 | 16 | 0.765 | −3 | 15 | 0.848 | −03 | 20 | 0.869 | −0.010 | 0.023 | 0.650 |
| amPFC | 19 | 25 | 0.463 | −43 | 34 | 0.206 | −13 | 17 | 0.451 | −36 | 29 | 0.221 | −0.021 | 0.028 | 0.444 |
| dmPFC | 16 | 15 | 0.312 | −27 | 28 | −0.334 | −21 | 9 | **0.020** | −27 | 13 | **0.040** | −0.076 | 0.030 | **0.011** |
| vmPFC | 26 | 44 | 0.562 | −50 | 53 | 0.350 | −10 | 25 | 0.686 | 0—50 | 48 | 0.300 | −0.006 | 0.041 | 0.875 |

Regions-of-interest (ROI) are defined in Fig. 1c. Significant p-values are bolded (p < 0.05, uncorrected, two-tailed). Linear mixed-effect modeling (LMEM) was used to analyze single-trial activation metrics (see Fig. 1e) from mentalizing-active ROI sites (see Fig. 3a). LMEMs controlled for stimulus visual dissimilarity (see Supplementary Methods) and accounted for site- and participant-related heterogeneity by nesting effects within Site and Participant. Separate LMEMs were used per ROI and activation metric (full details in Methods). Type$_{Other-Self}$ indicates mentalizing type differences, controlled for RT$_{Behav}$ (Fig. 5c–e). RT$_{Behav}$ indicates behavioral response time effects, controlled for mentalizing type differences, controlled for RT$_{Behav}$ (task in Fig. 1a). N per ROI (trials x sites): Visual = 8453, TPJ = 3189, ATL = 3046, PMC = 4301, amPFC = 2012, dmPFC = 3438, vmPFC = 1060. Abbreviations: ms = millisecond, z = z-score, b = slope estimate, SE = standard error of b, Visual = visual cortex, ATL = anterior temporal lobe, TPJ = temporoparietal junction, PMC = posteromedial cortex, amPFC = anteromedial prefrontal cortex, dmPFC = dorsomedial prefrontal cortex, vmPFC = ventromedial prefrontal cortex. Source data are provided as a Source Data file.

arithmetic (cognitive task) began to diverge. Specifically, in temporoparietal DMN regions, self- and other-mentalizing primarily evoked activations, while arithmetic evoked interdigitated activations and deactivations. From 500–750 ms, mentalizing activations began to encompass mPFC, especially for self-mentalizing. In contrast, arithmetic evoked mPFC deactivations, which continued for all successive time windows. From 750–1000 ms, both mentalizing types evoked similar mPFC activations. From 1000–2000 ms, other-mentalizing evoked more sustained activations than self-mentalizing, particularly in mPFC.

**Controlling for stimulus visual dissimilarity (VD).** To distinguish mentalizing-related neuronal effects (mentalizing type, $RT_{Behav}$, and $Choice_{Behav}$) from stimulus VD (e.g., prompt length), we used computer vision[36,37] (see Supplementary Methods). After controlling for VD, mentalizing-related effects were still significant in all DMN ROIs (Figs. 4d, e and 5c–e and Table 1). However, in visual cortex, marked self/other differences in peak power became nonsignificant (Fig. 5e). In sum, mentalizing-related effects in DMN ROIs were not explained by prompt length and other visual features.

**Behavioral results.** To confirm that participants performed mentalizing for task prompts, we analyzed $Choice_{Behav}$ for self/other biases towards positive or negative affective traits (Supplementary Fig. 4c) using logistic mixed-effects classification (nested within Trait within Participant). Overall, we found a greater probability of 'true' choices for positive versus negative traits (odds ratio = 2.53, $b_{1471} = 0.931 \pm 0.069$, $p = 1.04e{-}39$). This bias was stronger during self-mentalizing versus other-mentalizing (odds ratio = 1.38, $b_{1471} = 0.326 \pm 0.064$, $p = 4.56e{-}7$). These results indicate that participants engaged the mentalistic content in task prompts with a self-positivity bias, which is a canonical feature of mentalizing[38–40]. Moreover, high accuracy in arithmetic trials (median = 92.5%) suggests effortful attention to task prompts (Supplementary Fig. 4b).

To determine whether $RT_{Behav}$ varied by Mentalizing Type, $Choice_{Behav}$, and stimulus visual features (VD1 and VD2; see Supplementary Results), we used a LMEM nested within Participant (Supplementary Fig. 4a). We found faster $RT_{Behav}$ for self-mentalizing ($M = 2406 \pm 49$ ms) versus other-mentalizing ($M = 2797 \pm 52$ ms; $t_{1646} = 2.85$, $p = 0.004$). Additionally, $RT_{Behav}$ was faster for 'true' choices ($M = 2482 \pm 59$ ms) versus 'false' choices ($M = 2807 \pm 45$ ms; $t_{1646} = 2.16$, $p = 0.031$). However, the Mentalizing Type x $Choice_{Behav}$ interaction was nonsignificant ($t_{1646} = 1.47$, $p = 0.140$). Unsurprisingly, longer prompt lengths (VD1) elicited slower $RT_{Behav}$ ($t_{1646} = 4.85$, $p = 1.36e{-}6$). Meanwhile, remaining visual features (VD2) evoked nonsignificant $RT_{Behav}$ effects ($t_{1646} = -0.25$, $p = 0.803$). In sum, mentalizing type, $Choice_{Behav}$, and VD1 had significant and dissociable effects on $RT_{Behav}$.

## Discussion

Using electrocorticography (ECoG), we probed the neurocognitive substrates of mentalizing at the level of neuronal populations. We found that mentalizing about the self and others recruited near-identical cortical sites (Fig. 5a, b) in a common spatiotemporal sequence (Figs. 5c and 6). Within our ROIs, activations began in visual cortex, followed by temporoparietal DMN regions (TPJ, ATL, and PMC), and lastly in mPFC regions (amPFC, dmPFC, and vmPFC; Fig. 3e, f). Critically, regions with later activations exhibited greater functional specialization for mentalizing as measured by three metrics: functional specificity for mentalizing versus arithmetic (Figs. 3c, d and 4b), self/other differentiation in activation timing (Fig. 5c, d), and temporal

associations with behavioral responses (Fig. 4d and Table 1). Taken together, these results reveal a common neurocognitive sequence[28–31] for self- and other-mentalizing, beginning in visual cortex (low specialization), ascending through temporoparietal DMN areas (intermediate specialization), then reaching its apex in mPFC regions (high specialization).

Our results are consistent with gradient-based models of brain function, which posit that concrete sensorimotor processing in unimodal regions (e.g., visual cortex) gradually yields to increasingly abstract and inferential processing in 'high-level' transmodal regions like mPFC[41,42]. We found that the strength of self/other differences in activation timing increased along a gradient from visual cortex to mPFC. Specifically, other-mentalizing evoked slower (Fig. 5c) and lengthier (Fig. 5d) activations than self-mentalizing throughout successive DMN ROIs. These self/other functional differences corresponded with self/other differences in $RT_{Behav}$ (Supplementary Fig. 4), although the two were often dissociable (Table 1). Thus, perhaps because we know ourselves better than others, other-mentalizing may require lengthier processing at more abstract and inferential levels of representation, ultimately resulting in slower behavioral responses.

What might our results imply about extant fMRI findings? Hundreds of fMRI studies consistently suggest that: (1) TPJ and dmPFC are most crucial for mentalizing[6,8,11,12,43–46], and (2) dmPFC is selective for thinking about others over oneself[32–35]. However, when examined with ECoG, we found that both pieces of received wisdom are not what they seem. Below, we discuss both issues before moving onto our peculiar vmPFC results, and then conclude with systems-level discussion.

Unsurprisingly, we found that DMN regions such as TPJ and dmPFC contained higher proportions of 'mentalizing-specific' sites (i.e., mentalizing-active but not cognitive-active) relative to the whole-brain average (Fig. 3c, d). The spatial distribution of mentalizing-specific sites roughly resembles the 'mentalizing network' reported in countless fMRI studies[4,6,8–11,47]. However, our DMN ROIs were by no means functionally homogenous. Relative to other ROIs, TPJ and dmPFC activity best predicted $RT_{Behav}$ (Fig. 4d, e and Table 1), supporting the notion that both regions are most crucial for mentalizing performance[6,12,43,45,46,48,49].

We also found numerous functional distinctions between TPJ and dmPFC, which is surprising given their remarkably similar functional profiles in fMRI literature[4,6,8–11,47,50]. Specifically, we found that TPJ produced earlier activations (Fig. 3e, f) that were notably coactive for mentalizing and arithmetic (cognitive task; Fig. 3c, d). Indeed, the onsets of TPJ activations were the earliest of any DMN ROI. In contrast, dmPFC produced significantly later activations (Fig. 3e, f) that were overwhelmingly mentalizing-specific (Fig. 3c, d), indicating that dmPFC sits at a higher level of mentalizing's neurocognitive sequence than TPJ. Moreover, aggregate ROI analyses revealed no significant self/other differences in TPJ (Table 1), while dmPFC featured robust self/other timing differences (Fig. 5c, d), suggesting that dmPFC is more sensitive to variation in mentalistic content. In terms of behavior, although TPJ and dmPFC best predicted $RT_{Behav}$, dmPFC was the only ROI that predicted $Choice_{Behav}$ (choosing 'true' or 'false' for task prompts; Table 1), implying that dmPFC instantiates more aspects of mentalistic behavior than TPJ. Strikingly, unlike TPJ, dmPFC activation offsets closely preceded behavioral responses (within $124 \pm 7$ ms; Fig. 4c), suggesting that dmPFC is more deeply involved in the final stages of mentalistic reasoning. Taken together, while TPJ and dmPFC are both clearly crucial for mentalizing performance, dmPFC appears more specialized for mentalizing itself.

Given the marked functional differentiation between TPJ and dmPFC, what specific neurocognitive roles might they play in

mentalizing? In social neuroscience, TPJ is often considered a functionally-specific locus for explicit belief reasoning[45,46,51,52]. Yet here, TPJ was less functionally specialized relative to dmPFC (Figs. 3c–f, 4 and 5c, d). To explain this discrepancy, we suggest that TPJ provides crucial antecedents for explicit belief reasoning in dmPFC. Given TPJ's central role in automatic evaluations of thematic semantics[53–61], we propose that TPJ automatically represents integrative psycho-semantic models of exemplar contexts for a given inference. In simpler terms, TPJ may help us 'see' the psycho-semantic gestalt of a given situation[62]. Accordingly, tasks that 'show' concrete mentalistic content (e.g., social animations) often recruit TPJ but not dmPFC, while tasks that require mentalistic logical inferences (e.g., false belief) often recruit dmPFC in addition to TPJ[10–12]. Thus, when mentalistic content feels 'seeable' from perceptual processing, TPJ could generate mentalistic inferences without explicit belief reasoning. Indeed, work on implicit and spontaneous mentalizing show that TPJ can encode an actor's beliefs without any explicit reasoning[16,63–71]. Taken together, TPJ may implicitly set the psycho-semantic stage for explicit belief reasoning that occurs later in dmPFC when necessary (e.g., our trait judgment task; Fig. 1a).

The dmPFC may be well-suited for explicit belief reasoning[72–75]. We found substantial concurrent activation between dmPFC and all other ROIs (Figs. 2, 3e and 6), suggesting that dmPFC could work iteratively with lower-level regions to refine what is 'seen', thus providing dmPFC with increasingly-useful inputs from which to draw better inferences[76,77]. Moreover, studies on strategic reasoning show that dmPFC can arbitrate between multiple mental models[78] and prospective choices[79] by simultaneously evaluating multiple possibilities[80] through 'fuzzy' propositional reasoning[77,81]. As such, dmPFC may arbitrate between multiple exemplar contexts to help extract the most relevant and enduring semantic features for a given psychological inference. Taken together, dmPFC may integrate and refine representations throughout mentalizing's neurocognitive pathway to strategically reason about minds.

As for mentalizing about the self or others, fMRI studies routinely suggest that dmPFC is 'other-selective'[32–35]. What underlying neuronal population dynamics could result in stronger hemodynamic responses for one mentalizing type over another? The standard assumption would be that the magnitude (i.e., intensity) of neuronal activations differs across mentalizing type. This might be seen in aggregate ROI activity, or perhaps across individual ROI sites. We tested both possibilities by examining self/other differences in activation magnitude (HFB peak power). Unexpectedly, aggregate ROI analysis revealed that dmPFC produced nonsignificant self/other differences in peak power (Fig. 5e and Table 1). Similarly, single-site analysis showed that most dmPFC sites had nonsignificant self/other differences in peak power (86%), while the remaining 'self-greater' and 'other-greater' sites did not significantly outnumber each other (Fig. 5a, b). Strikingly, dmPFC did not contain sites that only activated for one mentalizing type. In sum, both mentalizing types recruited identical dmPFC sites (100% overlap) at equivalent intensities, which appears inconsistent with numerous fMRI reports of 'other-selective' dmPFC responses.

We instead found robust self/other differences in the timing of dmPFC activations. Specifically, other-mentalizing evoked slower and lengthier activations compared to self-mentalizing (Fig. 5c, d). In other words, dmPFC activity remained significantly above baseline for longer during other-mentalizing (see Figs. 2e and 6a). This suggests a different account of why dmPFC produces stronger hemodynamic responses for mentalizing about others over oneself. The typical story is that dmPFC is highly specialized for thinking about other people's minds[32–35]. Alternatively,

dmPFC could be sensitive to the inherently greater difficulty of other-mentalizing, which may necessitate additional processing cycles before completion. This additional processing may involve 'anchoring-and-adjustment'; the use of self-representations as an anchor from which to adjust other-mentalizing inferences – a function strongly linked with dmPFC and its role in strategic reasoning[75,82,83]. Self-mentalizing may be simplified by the rich compendium of accessible information we have about ourselves, thus resulting in brief but equally intense processing. Given that standard fMRI analysis does not distinguish activation intensity from activation duration, it appears that the latter has been mistaken for the former – though we cannot exclude the possibility of confounds related to differences between ECoG and fMRI[26].

Our peculiar vmPFC results may shed light on schematic contributions to mentalizing. Although there is growing consensus that vmPFC is important for schema processing[32,84–88], our results appear contrary to this at first blush. We found that vmPFC produced the latest activations across ROIs (Fig. 3e, f), which contradicts numerous reports of early (<200 ms) schema-based 'gist' predictions in vmPFC[89–95]. Early vmPFC activation has even been observed during mentalizing[17]. Of note, many studies have observed 'double waves' of early and late vmPFC activity: early activity may reflect vision-based gist construal, while late activity may reflect elaborative situational construal[93,96–100]. There is some evidence of a 'double wave' in the present study, as three posterior vmPFC sites produced very early activations (<200 ms; Figs. 2k and 4a) that met outlier exclusion criteria for ROI analyses. These early vmPFC sites produced equivalent activations for all task conditions (Figs. 2k, 3c and 5a), aligning with characterizations of rapid magnocellular gist processing, which likely cannot discern our alphanumeric stimuli[91,101]. In contrast, late-onset vmPFC sites produced longer activations for other- versus self-mentalizing (Figs. 2f and 5c, d), perhaps reflecting other-mentalizing's greater reliance on schematic feedback, especially in our trait judgment task[84,102–104]. Nevertheless, self- and other-mentalizing recruited near-identical vmPFC sites (91% overlap; Fig. 5a, b), suggesting common schematic underpinnings[32,84,105]. Taken together, we propose that vmPFC provides schematic contributions to mentalizing in two ways: rapid predictive processing from coarse visual afferents, followed by slow situational processing involving schematic feedback and integration across DMN.

At the systems level, we found complex and hierarchical processing dynamics across mentalizing's putative neurocognitive pathway. We observed consistent trial-by-trial propagation of activation onsets across ROIs (Supplementary Fig. 3), portraying an initial 'feedforward sweep' of coarser processing[28,106,107] along the pathway. Indeed, onset latencies were insensitive to self/other differences (Table 1). Onsets were followed by considerable concurrent activations across all ROIs (Figs. 2, 3e, 5c, 6 and S2), suggesting that distinct pathway regions largely work together (i.e., recurrent processing[28,108,109]) within an overarching processing sequence. Recurrent processing may be crucial for self/other differentiation, as self/other differences in aggregate ROI activity did not reach significance until concurrent activation was achieved across all ROIs (e.g., activation peaks and offsets; Fig. 5). Taken together, mentalizing may be supported by a brief initial 'feedforward sweep' of coarser processing along the pathway, followed by substantial recurrent processing that may integrate and refine representations across pathway regions. These dynamics could obscure cross-regional functional distinctions in fMRI studies. However, further research involving connectivity analyses are needed for more conclusive claims.

Temporoparietal DMN regions (tpDMN; TPJ, ATL, and PMC) may help integrate representations throughout mentalizing's neurocognitive pathway. Classification analysis revealed two

distinct functional types (Fig. 4b) that were anatomically interdigitated in tpDMN ROIs: sites with earlier non-specific activations, and sites with later mentalizing-specific activations (Fig. 3c). Intriguingly, non-specific tpDMN sites often coactivated with lower-level regions like visual cortex (e.g., Fig. 2b–d), suggesting attunement to lower-level afferents. Meanwhile, mentalizing-specific tpDMN sites often coactivated with higher-level regions like mPFC (e.g., Fig. 2e, f, h, i), suggesting attunement to higher-level feedback. Critically, we also found lengthy concurrent activations across site types and ROIs (Figs. 2 and 6), which could recurrently integrate low- and high-level representations[73,76,77,110]. Indeed, we found that visual and mentalistic representations were simultaneously encoded in all tpDMN ROIs (Supplementary Table 1). Taken together, we propose that interactions between distinct interdigitated neuronal populations in tpDMN[111–113] may help integrate and refine distant representations across mentalizing's neurocognitive pathway.

This study is not without confounds and limitations. Some of these limitations are inherent to ECoG: the use of participants with epilepsy, inconsistent brain coverage across participants, and sampling bias for cortical gyri[26]. Although these limitations were mitigated to the best of our ability (see Methods), they cannot be completely ameliorated. Thus, our ECoG findings could be corroborated by examining healthy participants with recent advances in source-space EEG/MEG, such as ultra-high density EEG[114], optically-pumped MEG[115], and laminar source localization[116]. Another important confound was the sparse right-hemisphere coverage of our cohort, which may limit the interpretability of our ATL and TPJ results[117]. Nonetheless, our few right-hemisphere sites appear functionally analogous to their left-hemisphere homologs (Supplementary Figs. 6–9). A task-related limitation was the short pre-stimulus baseline (Fig. 1a), which sometimes contained residual activity from prior trials, likely resulting in artifactual 'deactivations' in somatomotor sites (Fig. 6a). Another task-related confound was greater prompt length for other-mentalizing (e.g., "My neighbor is…") versus self-mentalizing (e.g., "I am…"; Supplementary Table 4), which we controlled for using computer vision[36,37] (see Supplementary Methods). Relatedly, mentalizing-specificity (Fig. 3c, d) could arise from differences between sentences and arithmetic equations. However, mentalizing-specific sites were not concentrated in reading-related regions[118–121], but rather in the 'mentalizing network' reported by countless fMRI studies[12].

Distributed hierarchical processing is a central organizing principle of neurocognitive systems[28,31,42,109]. Characterizing such hierarchies has enabled incisive neuromechanistic accounts of many psychological functions[30,122]. Here we provide a comprehensive electrophysiological exploration of the human social brain, revealing that mentalizing is characterized by complex and hierarchical neurocognitive dynamics at millisecond, millimeter, and cross-regional scales. While many questions remain, our findings contribute to a solid foundation upon which more conclusive neurocognitive accounts of mentalizing can be built.

## Methods

All research activities herein were conducted in accordance with protocol approved by the Stanford Institutional Review Board for human experimentation. All computational procedures and analyses herein were implemented in MATLAB[123] unless otherwise specified.

**Participants**. We employed a cohort of sixteen human participants while they underwent neurosurgical treatment for drug-resistant epilepsy (demographics in Supplementary Table 2). Each participant provided written informed consent in accordance with the Stanford Institutional Review Board. Participants were not compensated as per Stanford Institutional Review Board guidelines for inpatient clinical research.

As part of their presurgical evaluation, participants were implanted with ECoG at Stanford University Medical Center. The anatomical placement of electrode sites was determined according to each participant's clinical needs. Participants were included in this study's cohort if they had electrode coverage in key DMN regions: mPFC, PMC, TPJ, and ATL. Each participant was monitored in hospital for six to ten days prior to surgery, during which the study was conducted.

**Behavioral task**. ECoG data was recorded while participants performed an event-related behavioral task with six conditions (trial types; Fig. 1a and Supplementary Table 4). Five of these conditions required true/false responses to written prompts, while one condition consisted of cued rest. Two conditions featured mentalizing prompts, either about oneself (e.g., "I am honest") or others (e.g., "My neighbor is honest"). Participants were instructed to select a single neighbor (current or past) as the target for other-mentalizing. Cognitive task trials consisted of basic arithmetic (e.g., "9 + 86 = 95"). Two conditions featured memory-related prompts: episodic (e.g., "I ate candy yesterday") and self-semantic (e.g., "I eat a lot of candy"). The cued rest condition required no response and displayed a fixation crosshair for 5-10 seconds. The memory and rest conditions were not relevant to the current analyses and have been reported elsewhere[112,113,124]. Stimuli were presented in random order and were self-paced, advancing to the next trial after the participants' response, or up to 15 seconds if no response. The inter-trial interval (ITI) occurred -200-0 ms before each trial. The experiment was broken into two separate runs (mean run duration=12.50 ± 1.64 min). Participants were allowed a short break between behavioral runs. On average, each run featured 25 trials of each sentence condition, 40 cognitive trials, and 36 rest trials. Each non-rest trial contained unique prompts; prompts were not repeated within participant. Responses were made via a handheld keypad using either the '1' (true) or '2' (false) key. Participants were instructed to perform the task as accurately and as quickly as possible. All stimuli were presented in white font on a black background (1200 × 800 pixels) using Psychophysics Toolbox 3[125].

For all analyses, trials were excluded if meeting any of the following criteria: high-frequency epileptic oscillations, no behavioral responses, irrelevant button presses, or $RT_{Behav}$ under 400 ms.

**Electrocorticography data acquisition**. ECoG recordings were obtained via 2125 subdural electrodes (Fig. 1b). Electrodes (platinum plates with diameter of 1.2–2.3 mm) were implanted subdurally onto the cortical surface in grids or strips with center-to-center interelectrode spacing of 4-10 mm (Adtech Medical Instruments). Electrodes were connected to a multichannel recording system (Nihon Kohden; Tucker Davis Technologies) with sampling rate of 1000 Hz or above. Anatomical data was acquired using a GE 3-Tesla SIGNA Magnetic Resonance Imaging (MRI) scanner at Stanford University. A T1-weighted anterior-posterior commissure-aligned pulse sequence was used. T1 data was resampled to 1 mm isotropic voxels, then segmented to distinguish gray and white matter using FreeSurfer[126]. To facilitate electrode localization, postimplant computerized tomography (CT) scans were coregistered to the preoperative MRI anatomical brain volume[127]. For each participant, electrodes sites were localized in BioImage Suite[128] and displayed on the participants' own reconstructed 3D cortical surface using the iELVis toolbox[129]. Electrode positions were corrected for post-implantation brain shift, allowing for the accurate anatomical localization of electrodes sites[130].

**Defining regions of interest (ROIs) and brain networks**. Each participant's native-space cortical surface reconstruction (e.g., Supplementary Fig. 2) was used to classify electrode sites into a priori ROIs that are strongly implicated in mentalizing, with visual cortex included as a control ROI (Fig. 1c). Standardized brain-based parcellation was avoided due to known transformation inconsistencies in ECoG[129]. ROIs were defined through FreeSurfer cortical parcellation combined with visual inspection of anatomical landmarks. The ROI for 'visual cortex' consisted of occipital cortex, lingual gyrus, posterior fusiform gyrus, and posterior inferotemporal cortex. The 'ATL' ROI consisted of a bilateral anterior subregion of temporal cortex with precentral sulcus as the posterior bound, comprising the temporal poles and adjacent sections of entorhinal cortex and superior, middle, and inferior temporal sulci/gyri. The 'TPJ' ROI was a bilateral posterior subregion of inferior parietal lobule with lateral sulcus as the anterior bound, comprising angular gyrus and adjacent sections of supramarginal gyrus and superior temporal sulcus/gyrus. The 'PMC' ROI consisted of precuneus, posterior cingulate, and retrosplenial cortex. The 'amPFC' ROI was an mPFC subregion bounded between the ventral and dorsal reaches of corpus callosum. The 'dmPFC' ROI was a mPFC subregion ventrally bounded by the amPFC ROI and posteriorly bounded by the callosal rostrum. The 'vmPFC' ROI was an mPFC subregion dorsally bounded by the amPFC ROI and posteriorly bounded by the callosal rostrum, including the medial orbitofrontal surface.

**ECoG preprocessing**. Preprocessing was performed on a single-site/single-participant basis using custom routines (see Code Availability). First, data were notch filtered for power-line noise (57–63 Hz) and harmonics (117–123 Hz, 177–183 Hz). Electrode sites were discarded from further analyses if they were marked as pathological or 'noisy' by postclinical evaluation. The data was then rereferenced by

subtracting the mean signal of the remaining electrodes from each electrode's signal. The rereferenced data underwent time-frequency decomposition into 4–200 Hz spectra in 1–10 Hz bands using 5-cycle Morlet wavelet transforms. The power of the signal in each frequency band was z-transformed across time; this helps correct the 1/frequency decay of neurophysiological signals and improves interpretability. Data was then epoched into trials that were time-locked to stimulus presentation, ranging from 200 ms pre-stimulus to 5000 ms post-stimulus. For each trial and frequency band, baseline correction was performed by subtracting the mean power across the pre-stimulus baseline period ($-200$–0 ms) from all timepoints within a trial. To reconstruct the high-frequency broadband (HFB) signal, the primary signal of interest, z-transformed power of frequency bands within 70–180 Hz were averaged to produce a single HFB timecourse per electrode. Lastly, HFB timecourses from each electrode were low-pass filtered with a gaussian window (width = 50 ms) for further analysis. Trials were rejected from further analyses if epileptic high-frequency oscillations were observed. Within epochs, timepoints were discarded after presentation of the next trial's stimulus (i.e., timepoints after $RT_{Behav} + 200$ ms ITI).

**Statistics.** Statistical analyses described below were implemented in MATLAB Statistics and Machine Learning Toolbox[123]. Data was visualized using the GRAMM toolbox[131]. All statistical tests were two-tailed. All multiple comparisons corrections maintained the False Discovery Rate (FDR) under 0.05 through the Benjamini-Yekutieli procedure for data with any dependence structure[132], with p-values adjusted accordingly ($p_{FDR}$).

We primarily used mixed models due to their design flexibility and robustness to sampling heterogeneity, multicollinearity, and statistical confounds[133]. Specifically, we employed linear mixed-effects modeling (LMEM) for continuous outcomes and logistic mixed-effects classification (LMEC) for binary outcomes. To minimize estimation bias, mixed models used restricted maximum likelihood-based estimators[133]. To account for full data dependence structure with reduced bias and assumptions, mixed models unconstrained variance-covariance matrices with log-Cholesky parametrization[134]. To account for heterogenous variances across mixed model terms, Satterthwaite approximation for degrees of freedom (effective $DF$) was used[135]. To rectify violations of assumptions and overparameterization in mixed models, we evaluated objective function Hessian matrices and ensured positive definiteness[136]. Additionally, non-mixed models were used when specified below.

**Within-site analyses.** Within-site analyses (Fig. 1d–f) were performed to provide the bases for the primary multi-site analyses. The dependent variable for within-site analyses was z-scored HFB power across timepoints and trials (Fig. 1d). Trials were excluded if containing high-frequency epileptic oscillations, no behavioral responses, irrelevant button presses, or $RT_{Behav}$ under 400 ms. Timepoint-by-timepoint observations met outlier exclusion criteria if HFB power exceeded three median absolute deviations from other observations of the same timepoint and task condition.

**Trial-averaged analysis (within-site).** To identify sites with significant evoked HFB responses for each task condition ($p_{FDR} < 0.05$, corrected across timepoints and sites), we used LMEMs (Fig. 1f). The intercept (null distribution) consisted of timepoints within the pre-stimulus baseline ($-200$–0 ms). Each peri-stimulus timepoint (0–5000 ms) was represented as a separate dummy variable. The intercept was nested within trial to account for trial-specific variance. This LMEM specification estimates mean timecourses of z-scored HFB power ($\beta$) for each task condition. To dampen spikes and other noise, timepoints were not considered significant unless $p_{FDR} < 0.05$ was maintained for 50 ms consecutively. For each task condition, sites were considered 'active' or 'deactive' if evoked HFB power was significantly higher or lower, respectively, than pre-stimulus baseline; if sites produced both, the polarity of the greatest deflection was used. Sites with non-significant differences from baseline were considered 'nonresponsive'.

**Single-trial analysis (within-site).** To identify timepoints with significant evoked HFB activations for each trial (Fig. 1e), timepoints between stimulus onset and the forthcoming trial ($RT_{Behav} + 200$ ms) were run through a sliding window test (width = $\pm 10$ ms). Observations (z-scored HFB power) in each sliding window were tested against observations from the pre-stimulus baseline via two-sample Welch's $t$-tests, which accounted for unequal variances and sample sizes. This analysis identified timepoints within individual trials that featured significant stimulus-evoked responses ($p_{FDR} < 0.05$, corrected across timepoints, trials, and sites) relative to the pre-stimulus baseline preceding each trial. To dampen spikes and other noise, timepoints were not considered significant unless $p_{FDR} < 0.05$ was maintained for 50 ms consecutively.

Single-trial analysis provided five key metrics of HFB activity (Fig. 1e). *Onset latency* is the earliest timepoint with significant activation (green squares). *Peak latency* and *peak power* are the timepoint and magnitude, respectively, of the strongest activation (white squares). *Offset latency* is the latest timepoint with significant activation (red squares). *Duration* is the total number of timepoints with significant activations (brown areas).

**Multi-site analyses.** Multi-site analyses used results from within-site analyses as response measures. Outliers were identified with respect to the two-dimensional distance between stimulus onset and $RT_{Behav}$ using bisquare robust regression[106]. For each site and single-trial HFB metric, observations were considered outliers and discarded if residuals were greater than three median absolute deviations (MAD). For ROI sites, this method was applied a second time using observations from all mentalizing-active sites within each ROI. Sites were excluded from ROI analyses if over 50% of its observations exceeded three MAD. Of all ROI sites, only three sites in vmPFC were excluded.

**Functional specificity and selectivity (multi-site).** Functional specificity (Fig. 3c, d) of each site was determined by comparing mentalizing (collapsed across self/other) and arithmetic (cognitive task) results from trial-averaged analysis, along with direct comparisons of single-trial peak power (including trials with nonsignificant activations) for mentalizing versus arithmetic using bisquare robust regression. Specifically, sites were considered 'mentalizing-specific' if they produced (1) significant trial-averaged activations for mentalizing but not arithmetic ($p_{FDR} < 0.05$, corrected across sites and timepoints), and (2) produced greater peak power for mentalizing over arithmetic ($p_{FDR} < 0.05$, corrected across sites). Sites with significant trial-averaged coactivations for mentalizing and arithmetic were considered 'non-specific', regardless of peak power differences. All mentalizing-active sites with nonsignificant mentalizing/arithmetic peak power differences were labeled as 'non-specific' (Mentalizing active = Cognitive active).

Self/other selectivity (Fig. 5a, b) of each site was determined by trial-averaged results and direct comparisons of single-trial peak power (including trials with nonsignificant activations) for self- versus other-mentalizing using bisquare robust regression. Sites were considered 'self-only' or 'other-only' (not considering cognitive task) if they only produced (1) significant trial-averaged activations for only one mentalizing type ($p_{FDR} < 0.05$, corrected across sites and timepoints) and (2) produced greater peak power for that mentalizing type over the other ($p_{FDR} < 0.05$, corrected across sites). Sites that activated for both mentalizing types were labeled by self/other differences in peak power: 'self-greater' (Self > Other), 'other-greater' (Other > Self), and 'non-selective' (Self = Other). Sites that activated for only one mentalizing type but had nonsignificant self/other differences in peak power were considered 'non-selective' (Self = Other). In sum, self-only and self-greater sites were considered 'self-selective', while other-only and other-greater sites were considered 'other-selective'.

**Aggregate ROI analyses (multi-site).** To reveal the spatiotemporal dynamics of mentalizing processing, we analyzed single-trial HFB metrics from mentalizing-active ROI sites. Each ROI and HFB metric was analyzed using separate LMEMs. All LMEMs were nested within Participant, which accounted for within- and between-participant heterogeneity through unconstrained variance/covariance matrices across within-participant random effects. LMEMs that included Site or Trial as nesting factor likewise accounted for site- and trial-related heterogeneity, respectively. Stimulus visual dissimilarity was controlled for by including two visual dimensions (VDs) as random effects specified below; VDs were derived from a popular computer vision model based on the ventral visual stream[36,37] (see Supplementary Methods). In addition to outlier exclusion criteria in the *multi-site analyses* section above, trials with $RT_{Behav}$ over 5000 ms were excluded given our 5000 ms epoch lengths.

LMEMs for self/other differences (Fig. 5c–e and Table 1) represented mentalizing type as a fixed and random effect. We also included random effects for $RT_{Behav}$ and VDs, which controlled for $RT_{Behav}$ and stimulus visual dissimilarity. All random effects were nested within Site and Participant to account for site- and participant-related heterogeneity.

LMEMs for neurobehavioral associations (Fig. 4d, e and Table 1) simultaneously estimated $RT_{Behav}$ and $Choice_{Behav}$ (controlling for one another) as fixed and random effects, while VDs were specified as random effects. All random effects were nested within Site and Participant to account for site- and participant-related heterogeneity.

Pairwise ROI comparisons (Fig. 3f) were performed using LMEMs with ROI as a fixed and random effect nested within Trial within Participant, which estimated within-trial ROI differences and accounted for trial- and participant-related heterogeneity. Additionally, a random effect for $RT_{Behav}$ was nested within Trial within Participant to control for $RT_{Behav}$. Critically, each pairwise comparison was restricted to participants with mentalizing-active sites in both ROIs.

**Whole-brain HFB responses within time-windows (multi-site).** To provide a broad overview of the neuronal spatiotemporal dynamics evoked by each task condition, we performed whole-brain analysis of HFB responses within specific time-windows (Fig. 6a). For each site and task condition, separate LMEMs were used to analyze trial-by-trial HFB power (z-scored). The intercept (null distribution) consisted of observations within the pre-stimulus baseline ($-200$–0 ms). Dummy variables consisted of observations within each time window. All model terms were specified as fixed and random effects nested within Trial, which accounted for trial-related heterogeneity. This specification estimates HFB

responses evoked by each task condition within time windows ($p_{FDR} < 0.05$, corrected across sites and time windows).

**Grand-average ROI timecourses (multi-site)**. To summarize task-evoked neuronal dynamics within ROIs, we performed grand-averaged analysis (Fig. 6b–h) of trial-by-trial HFB timecourses (z-scored) from ROI sites that were identified as active or deactive for a given task condition by trial-averaged analysis (see Figs. 1f and 2). Separate LMEMs were used for each ROI and task condition. The intercept (null distribution) consisted of timepoints within the pre-stimulus baseline (−200–0 ms). Peri-stimulus timepoints (0–3000 ms) were represented as dummy variables. Model terms were nested within Site and Participant to account for site- and participant-related heterogeneity. This specification estimates grand-average HFB timecourses evoked by each task condition within ROIs ($p_{FDR} < 0.05$, corrected across ROIs and timepoints).

**Reporting summary**. Further information on research design is available in the Nature Research Reporting Summary linked to this article.

## Data availability

Anonymized preprocessed ECoG data can be shared upon reasonable request to Kevin M. Tan (kevmtan@ucla.edu), subject to a data-sharing agreement between the requestor(s), study authors, and Stanford University. The data-sharing agreement will be tailored to the aims of the requestor(s). Source data are provided with this paper.

## Code availability

Custom code is available here: https://github.com/pinheirochagas/lbcn_preproc. Additional information, annotation, and organization of code used here can be provided upon request to Kevin M. Tan (kevmtan@ucla.edu).

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

## Acknowledgements

We are grateful for public financial support from the United States and Canada. This work was supported by National Institute of Neurological Disorders and Stroke Grant R01NS078396 (J.P.), National Institute of Mental Health Grant 1R01MH109954-01 (J.P.), National Science Foundation Grant BCS1358907 (J.P.), Department of Defense Grant 13RSA281 (M.D.L.), National Institute of Child Health and Human Development Postdoctoral Fellowship F32HD087028 (A.L.D.), Stanford University School of Medicine Medical Scholars Research Fellowship (K.C.R.F), Natural Sciences and Engineering Research Council of Canada Postdoctoral Fellowship (K.C.R.F.), and National Science Foundation Graduate Research Fellowship DGE-1650604 (K.M.T.). We thank the Laboratory of Behavioral and Cognitive Neuroscience for generously providing access and support for these data. We thank members of the Social Cognitive Neuroscience Lab for their continued support. We also thank Carolyn Parkinson for her continued support.

## Author contributions

J.P. and A.L.D. contributed to experimental design and data acquisition. J.P., A.L.D, P.P., and K.M.T. developed preprocessing and analysis tools. K.M.T. and M.D.L. conceptualized the analysis plan. K.M.T. performed formal data analysis. All authors contributed to writing of the manuscript.

## Competing interests

All authors declare no conflicts of interest.
