## [Peer Review File · Nature Communications]

Electrocorticographic evidence of a common neurocognitive sequence for mentalizing about the self and othersREVIEWER COMMENTS

Reviewer #1 (Remarks to the Author):

Here, Kevin Tan and group provide a highly interesting and timely study on the neural mechanisms that underlie mentalizing in humans and processes by which we distinguish self from other. To investigate the spatiotemporal dynamics of mentalizing, the authors recorded neuronal population activity using ECoG while subjects judged the traits of themselves and others. They concluded that “self- and other-mentalizing recruited near-identical neuronal populations in a common spatiotemporal sequence: activations were earliest in visual cortex, followed by temporoparietal DMN regions, and finally medial prefrontal cortex.” Regions that were activated later displayed greater functional specificity for mentalizing and had greater self-other differentiation. They also showed stronger associations with behavioral response times. Finally, when comparing self- vs other-mentalizing in the temporal domain, the latter evoked slower and lengthier activations across successive DMN regions, suggesting longer computation times needed for more abstract and inferential levels required for mentalizing about others.

Overall, this study aims to address an important question about the dynamics of mentalizing. It provides novel and excited findings, and I would support its publication. Below are points/questions though that I would hope can be addressed beforehand.

First, the authors define mentalizing-active sites in this study as those showing higher HFB power relative to pre-stimulus baseline: e.g., P5 L22-24. Later, they define ‘mentalizing-specific’ (light + dark turquoise) sites as those that were active for mentalizing but were nonresponsive for arithmetic. However, if I understand correctly, for their later temporal analysis of self vs. other they analyze the mentalizing-only sites and not mentalizing-specific sites (e.g., P8, L15-16). It is not clear to me, however, why they are not using mentalizing-specific sites for the analysis. My concern here is that the definition of mentalizing-only may be potentially confounded. All we know is that these sites get activated when presenting mentalizing stimuli (and possibly arithmetic stimuli) which are all visual stimuli. How do the authors rule out the possibility that the mentalizing-only activations are not more simply reflecting responses to any visual stimulus? To address this, I would consider re-analyzing the data using stricter criteria, e.g., using mentalizing-specific activation instead. It may also be helpful to evaluate the robustness of the results using other non-visual stimuli if possible.

Second, on a similar note, it may be helpful if the authors could use the cued rest condition in which they show a visual fixation point as the baseline for visual activation. Without subtracting the impact of visual stimuli, it is not clear how one can attribute the activity of mentalizing-only sites to actual mentalization.

Third, it was not clear to me how the authors confirmed that the subjects are paying attention? Are there behavioral metrics on a trial-by-trial basis that correlates to activity?

Fourth, it may be helpful to align the activity to the response time as well for control. It would also be helpful to provide additional information on length of the trials. For example, the authors mention as a caveat that the length of other mentalizing trials may be larger than those of the self-mentalizing trials but do not clearly state whether or how this impacts the results.

Finally, it would be helpful to provide additional examples of the stimuli themselves.

Reviewer #2 (Remarks to the Author):

Tan et al. present an ECoG-based study of self- and other-directed mentalizing. Their results help to validate years of fMRI and MEG/EEG findings on social cognition, using a method that provides a combination of spatial and temporal resolution unavailable to these noninvasive techniques. They

conclude that activity related to mentalizing progresses up a cortical hierarchy from visual cortex through the default network, ending in medial prefrontal cortex.

The findings presented in this paper are effectively unique – due to the challenges involved in ECoG – and highly valuable to the social neuroscience literature. Moreover, the experimental design, analyses, and results are clear and thorough. Although they do not diminish my enthusiasm for this work, I describe below a few issues that I think deserve further attention. I am confident that these can be addressed through relatively minor changes to the analyses.

1. The definition of “mentalizing-specific” sites is statistically problematic. These sites are defined as those that showed a statistically significant positive effect for the mentalizing condition (relative to baseline) but no statistically significant positive effect for the arithmetic control condition. However, such a difference in statistical significance is not, in itself, statistically significant. A mentalizing specific region could have nearly identical activity in response to mentalizing vs. arithmetic, as long as mentalizing fell just below $p = .05$, and arithmetic fell just above $p = .05$. As a result, it is quite possible that there were much larger mentalizing>arithmetic differences in so-called non-specific sites (where both conditions were significant relative to baseline) than in nominally mentalizing-specific sites defined in this manner. This seems inherently problematic for the interpretation of both mentalizing-specific and non-specific sites. I would suggest instead defining mentalizing-specific sites as those that show a significant increase in activity during mentalizing relative to both baseline and the arithmetic condition. This issue also applies to the definitions of “self-only” and “other-only” sites. In that case, however, these problematic labels are supplemented by self-greater, other-greater, and non-selective regions defined by directly contrasting the self and other conditions, mitigating the issue in that case.

2. The correlation between mentalizing specificity and onset latency has only five observations. The same is also true of the other ROI-level latency correlations. Despite the large magnitudes of the observed associations – and correspondingly significant p-values – I do not find these results convincing in their present form. Correlations over so few observations are extremely unstable, and liable to yield inflated estimates when significant. I would suggest disaggregating the data in some way(s) – for example, by sites within region, by participant, or by trial – and reassessing these associations using the large number of observations that such disaggregation would allow.

3. It is notable that vmPFC does not show greater activity for self > other, and if anything, trends in the other direction. This would seem to be at odds with a large number of fMRI findings indicating that this region is involved in self-relevant processing. Perhaps the present result is simply a false negative – we cannot know for certain. However, it would be interesting to hear consideration of this result in the discussion, and any potential explanation for the discrepancy between ECoG and fMRI in this regard.

Minor:

- Page 2, line 8: “higher...” higher implies a comparison, but no other modality is mentioned in this sentence or the preceding or following one. Implicitly the comparison is to fMRI, but that comparison doesn't become explicit until later.
- The bottom “latency” labels in figure 2 appear partially cut off.

Reviewer #3 (Remarks to the Author):

This paper addresses the important question of the neural mechanisms for reasoning about ourselves and others (social neuroscience). The authors studied the spatio-temporal patterns of activation of the human brain during self-mentalizing, other-mentalizing, and mental calculations using a series of true-false questions. They use a localizer task while recording intracranial EEG recordings to localize contacts selective for mentalizing trials and analyze their response latency (both based on high-frequency gamma band power, HFB).

Results: First, the proportion of electrodes selectively responding to mentalizing trials increases from posterior to anterior (across visual, TPJ, ATL, PMC, PFC). Second, similar for the onset latency (Fig. 3), establishing a sequence of activations anterior to posterior. Third, all areas examined contained a mixture of self, other and both conditions (Fig 4), with differences between self and other in terms of latency of peak and duration of activation.

Positive: The conclusions are based on a large dataset of high quality. The results are novel and offer a comprehensive large-scale mapping of the location and response latency of contacts carrying self/other selective HFB activity. This task is much studied in social neuroscience and thus of high relevance.

Negatives: Design caveats in the task pose numerous caveats on the interpretation. Also puzzling are several bold claims are made that are not supported by this rather simple localizer task. There is no relationship between the neural activity assessed and behavior in a given trial, leaving the behavioral relevance of the activity observed unclear.

Major issues:

1. Behavior and task.

First, there is no analysis of behavior. How do we know the patients were successful at mentalizing about themselves or others (and doing mental math) ? What are the distribution of RTs, responses etc ?

Second, the task seems not well designed to address the overall question. The three types of trials vary in several ways: reading words vs numbers, number of words, and RT. The fact that only the mentalizing conditions included words is a confound for the mentalizing vs cognitive contrasts – one could equally call this a “reading words vs numbers” contrast? Also at least in the example the number of words is higher for the ‘other mentalizing’ trials.

2. Relationship between RT and latency/power (Fig 4). First, these differences could be due to differences between the trials (task design, see above) rather than self vs other differences. RTs are longer in the other condition, so the effects seen in Fig 4c-e might be due to this. As controls, one would expect analysis where subsets of trials are selected where RT (and number of words, if that is possible) are matched, then examining whether the observed latency differences still exist. Using a mixed effect model to account for effects of RT does not solve this problem. Second, RT differences could be due to tuning to choices (true vs false). Are the proportion of true/false answers balanced between the self vs other groups? Third, the conclusion of 'stronger association with response times' with areas that respond later are not supported by Fig 4e - the associations seem similar in magnitude and this statement is based on a null result. Overall, Fig 4 was the least convincing part of the paper.

3. Within vs across patient differences. How are differences across patients taken into account? i.e. presumably some patients had TPJ electrodes, and others had vmPFC electrodes. Could the latency differences be due to generally slower processing in some vs other patients? The mixed effect models do not appear to take into account between-subject variance.

4. Claim of ‘near identical populations’ (made several times, including abstract). I cannot see how this is supported based on comparing gamma power. HFB is the sum of synaptic activity across a large number of neurons. This does not allow conclusion about which groups of neurons are activated. All that is supported is that gamma band power did not differ.

5. Introduction/discussion. The introduction and discussion is simplistic and does not reflect the physiological basis of what is measured – it is assumed that fMRI and EEG/MEG/ECOG measure the same thing, but with different temporal and spatial resolution. There is certainly more than the ‘limited temporal resolution of fMRI’ that precludes fMRI from revealing a ‘precise account of mentalizing and its accounts’. It would help the paper by properly framing the differences in what is measured here vs. what fMRI measures (metabolic activity).

6. Claim of a 'common neurocognitive pathway' that is 'hierarchical'. I cant see how these claims are supported. What is shown is an impressive and very interesting set of latency and HFB power tuning differences across cortex. But how does this show a pathway and a hierarchy? Seeing latency differences does not make something 'a pathway' - this does not show that these areas communicate with each other (pathway) nor that they form a hierarchy. Similar for the statement of 'ascending through' and similar in the discussion – I cannot see how the data support that. Trial-by-trial analysis of latency differences across simultaneously recorded contacts would be needed to support these claims.

Dear reviewers,

We are grateful for the interest in our manuscript and have made all possible efforts to ameliorate reviewer concerns. We also made a few additional changes not related to reviews or related to the concerns of all reviewers. These changes are listed here, prior to addressing each specific concern raised in the review process.

First, upon closer inspection of clinician notes, some sites were mislabeled as 'noisy', which has since been corrected. This affected only 16 of 555 ROI sites (<3%), resulting in minimal changes to the results.

Second, given overarching concerns about behavioral RTs, we discarded trials with RT over 5000 ms in analyses of HFB latencies/duration. 5000 ms was decided as the outlier cut-off during preprocessing, and HFB data beyond 5000 ms was not preprocessed due to added strain on limited computing resources. Arithmetic RTs exceeded 5000 ms with some regularity in some subjects, thus any mention of arithmetic HFB latencies was removed from the supplementary materials. These changes are reflected in Results (pg. 5) and Methods (pg. 26).

Relatedly, to examine HFB activations after behavioral RTs, we no longer discard single-trial HFB data after RTs, but instead after presentation of the next trial's stimulus. This provides 200 ms (ITI) of HFB data after RTs. This change is reflected in the Methods section and the caption for Fig. 1.

Third, we have also moved the all-ROI omnibus results from Table 1 to Table S1, since the omnibus results were not referenced in the main text, and do not add to the primary results. They may have also mistakenly given the impression that the within-ROI and pairwise-ROI results were just derived from the all-ROI statistical models, instead of being separate analyses.

Lastly, the summary analyses involving whole-brain time windows (Fig. 6a) and grand-average ROI timecourses (Fig. 6b-h) now use trial-by-trial observations, which should be more statistically sound than the previous method of averaging results from single-site analyses (see Methods, pg. 27-28).

Point-by-point responses to reviewer concerns are found below, with page breaks after each reviewer.

Sincerely,

Kevin Tan and co-authors

R1 opening: Here, Kevin Tan and group provide a highly interesting and timely study on the neural mechanisms that underlie mentalizing in humans and processes by which we distinguish self from other. To investigate the spatiotemporal dynamics of mentalizing, the authors recorded neuronal population activity using ECoG while subjects judged the traits of themselves and others. They concluded that “self- and other-mentalizing recruited near-identical neuronal populations in a common spatiotemporal sequence: activations were earliest in visual cortex, followed by temporoparietal DMN regions, and finally medial prefrontal cortex.” Regions that were activated later displayed greater functional specificity for mentalizing and had greater self-other differentiation. They also showed stronger associations with behavioral response times. Finally, when comparing self- vs other-mentalizing in the temporal domain, the latter evoked slower and lengthier activations across successive DMN regions, suggesting longer computation times needed for more abstract and inferential levels required for mentalizing about others. Overall, this study aims to address an important question about the dynamics of mentalizing. It provides novel and excited findings, and I would support its publication.

Response: We would like to give many thanks for your kind words and helpful suggestions! We hope we have ameliorated your concerns.

R1, Point 1: First, the authors define mentalizing-active sites in this study as those showing higher HFB power relative to pre-stimulus baseline: e.g., P5 L22-24. Later, they define ‘mentalizing-specific’ (light + dark turquoise) sites as those that were active for mentalizing but were nonresponsive for arithmetic. However, if I understand correctly, for their later temporal analysis of self vs. other they analyze the mentalizing-only sites and not mentalizing-specific sites (e.g., P8, L15-16). It is not clear to me, however, why they are not using mentalizing-specific sites for the analysis. My concern here is that the definition of mentalizing-only may be potentially confounded. All we know is that these sites get activated when presenting mentalizing stimuli (and possibly arithmetic stimuli) which are all visual stimuli. How do the authors rule out the possibility that the mentalizing-only activations are not more simply reflecting responses to any visual stimulus? To address this, I would consider re-analyzing the data using stricter criteria, e.g., using mentalizing-specific activation instead. It may also be helpful to evaluate the robustness of the results using other non-visual stimuli if possible.

Response: This is a fair concern, however, this study aimed to examine mentalizing and its non-mentalizing antecedents, including even visual processing. Thus, we analyzed all mentalizing-active sites. Moreover, visual cortex, ATL and TPJ contained <40% mentalizing-specific sites; using only mentalizing-specific sites would be highly detrimental to analyses of these ROIs.

However, we compared the onset latencies of mentalizing-specific and non-specific sites; onset latencies robustly predicted functional specificity (pg. 8 & Fig. 4b). We also added systems-level discussion on the possible roles and interactions of non-specific and mentalizing-specific sites in temporoparietal DMN regions (pg. 18-19):

“Classification analysis revealed two distinct functional types (Fig. 4b) that were anatomically interdigitated in tpDMN ROIs: sites with earlier non-specific activations, and sites with later mentalizing-specific activations (Fig. 3c). Intriguingly, non-specific tpDMN sites often coactivated with visual and attentional regions, while mentalizing-specific tpDMN sites often coactivated with mPFC (Figs. 6 & 2). Thus, we speculate that non-specific tpDMN sites may be more attuned to lower-level afferents, while mentalizing-specific tpDMN sites may be more

attuned to higher-level feedback. Critically, we also found lengthy concurrent activations across site types and ROIs (Figs. 6 & 2), which could integrate low- and high-level representations, such as visual features and mentalistic inferences.”

As for having a visual control, the pre-stimulus baseline (ITI) contained a fixation crosshair, which helps control for low-level visual effects. Unfortunately, we do not have non-visual stimuli available to use as a control; self-mentalizing, other-mentalizing, and arithmetic all used alphanumeric prompts.

Mentalizing-active sites, which formed the basis of ROI analyses, are defined as sites with any significant activation to mentalizing relative to the fixation pre-stimulus baseline. Mentalizing-specific sites should not have any arithmetic activations but can include arithmetic deactivations. Moreover, the criteria for ‘mentalizing-specific’ was made stricter in response to statistical concerns from another reviewer (pg. 6): “Sites were considered ‘mentalizing-specific’ they were 1) mentalizing-active but not arithmetic-active, and 2) produced significantly higher peak power for mentalizing over arithmetic.”

R1, Point 2: Second, on a similar note, it may be helpful if the authors could use the cued rest condition in which they show a visual fixation point as the baseline for visual activation. Without subtracting the impact of visual stimuli, it is not clear how one can attribute the activity of mentalizing-only sites to actual mentalization.

Response: The pre-stimulus baseline (ITI) contained the same fixation crosshair as the cued rest condition, and thus involves subtraction of visual stimuli. Using the pre-stimulus baseline instead of the cued rest condition leverages one of the primary benefits of ECoG (and EEG/MEG): the ability to use trial-by-trial baselines instead of averaged baseline and condition contrasts.

R1, Point 3: Third, it was not clear to me how the authors confirmed that the subjects are paying attention? Are there behavioral metrics on a trial-by-trial basis that correlates to activity?

Response: To highlight that we confirmed that subjects were paying attention, we have added these sentences to the Results (pg. 5) and Methods (pg. 20) sections: “For all analyses, trials were excluded if they met any of these criteria: irrelevant button presses, RT_{Behav} under 400 ms, or no behavioral responses.”

Trial-by-trial behavioral responses were correlated with HFB activity in Figs. 4c-e & Table 1. Moreover, behavioral responses were accounted for in analyses of cross-ROI latency differences (Fig. 3ef) and self/other differences in ROIs (Fig. 5c-e).

We have also changed “ RT_{Task} ” to “ RT_{Behav} ” in the manuscript to make clearer the behavioral aspect of the analyses.

R1, Point 4: Fourth, it may be helpful to align the activity to the response time as well for control. It would also be helpful to provide additional information on length of the trials. For example, the

authors mention as a caveat that the length of other mentalizing trials may be larger than those of the self-mentalizing trials but do not clearly state whether or how this impacts the results.

Response: We have added analysis of HFB offsets relative to behavioral RTs in Fig. 4c and Results (pg. 7): “We also performed post-hoc analysis of offset latencies relative to RT_{Behav} (Fig. 4c), revealing that mPFC activations more closely preceded RT_{Behav} than other ROIs combined ($b_{20204}=138$ ms, $SE=31$, $p=4.28e-6$).”

We also added the following to a discussion paragraph on self/other differences (pg.13):

“These self/other functional differences corresponded with self/other differences in RT_{Behav} (Fig. S4), though the two were still dissociable (Table 1). Thus, perhaps because we know ourselves better than others, other-mentalizing may require lengthier processing at more abstract and inferential levels of representation, ultimately resulting in slower behavioral responses. ”

R1, Point 5: Finally, it would be helpful to provide additional examples of the stimuli themselves.

Response: We have added many additional examples of the stimuli in Table S4.

R2 opening: Tan et al. present an ECoG-based study of self- and other-directed mentalizing. Their results help to validate years of fMRI and MEG/EEG findings on social cognition, using a method that provides a combination of spatial and temporal resolution unavailable to these noninvasive techniques. They conclude that activity related to mentalizing progresses up a cortical hierarchy from visual cortex through the default network, ending in medial prefrontal cortex.

The findings presented in this paper are effectively unique – due to the challenges involved in ECoG – and highly valuable to the social neuroscience literature. Moreover, the experimental design, analyses, and results are clear and thorough. Although they do not diminish my enthusiasm for this work, I describe below a few issues that I think deserve further attention. I am confident that these can be addressed through relatively minor changes to the analyses.

Response: Many thanks for the kind words and very useful comments! We hope we have addressed all your concerns.

R2, Point 1: The definition of “mentalizing-specific” sites is statistically problematic. These sites are defined as those that showed a statistically significant positive effect for the mentalizing condition (relative to baseline) but no statistically significant positive effect for the arithmetic control condition. However, such a difference in statistical significance is not, in itself, statistically significant. A mentalizing specific region could have nearly identical activity in response to mentalizing vs. arithmetic, as long as mentalizing fell just below $p = .05$, and arithmetic fell just above $p = .05$. As a result, it is quite possible that there were much larger mentalizing>arithmetic differences in so-called non-specific sites (where both conditions were significant relative to baseline) than in nominally mentalizing-specific sites defined in this manner. This seems inherently problematic for the interpretation of both mentalizing-specific and non-specific sites. I would suggest instead defining mentalizing-specific sites as those that show a significant increase in activity during mentalizing relative to both baseline and the arithmetic condition. This issue also applies to the definitions of “self-only” and “other-only” sites. In that case, however, these problematic labels are supplemented by self-greater, other-greater, and non-selective regions defined by directly contrasting the self and other conditions, mitigating the issue in that case.

Response: These are very helpful comments. We have added direct comparisons of HFB peak power across mentalizing and arithmetic in functional specificity analysis (Fig. 3cd), resulting in a stricter definition of mentalizing-specificity (pg. 6): “Sites were considered ‘mentalizing-specific’ (light + dark turquoise) if they were 1) mentalizing-active but not arithmetic-active, and 2) produced significantly higher peak power for mentalizing over arithmetic.”

We did not consider “Mentalizing-active > Arithmetic-active” sites as ‘mentalizing-specific’ because they still produced significant activations for arithmetic over baseline. Moreover, given the large number of visual cortex sites with this response profile, it may in part reflect low-level differences between sentences and numbers.

Similarly for self/other, we now have stricter criteria for ‘self-only’ and ‘other-only’ sites (pg. 9-10): “Sites were considered ‘self-only’ or ‘other-only’ if they 1) produced significant trial-averaged activations for only one mentalizing type, and 2) exhibited significantly greater peak power for that mentalizing type over another.”

This resulted in fewer self-only and other-only sites, including eliminating self-only sites in amPFC, which were spurious despite the literature.

Peak power comparisons in specificity & selectivity analysis now include trials with nonsignificant activations to further account for subthreshold activations. Moreover, these peak power comparisons now use robust regression instead of t-tests to help account for noise in peak power. These changes are reflected in the Results section (pg. 6 & 9) and Methods section (pg. 25).

R2, Point 2: The correlation between mentalizing specificity and onset latency has only five observations. The same is also true of the other ROI-level latency correlations. Despite the large magnitudes of the observed associations – and correspondingly significant p-values – I do not find these results convincing in their present form. Correlations over so few observations are extremely unstable, and liable to yield inflated estimates when significant. I would suggest disaggregating the data in some way(s) – for example, by sites within region, by participant, or by trial – and reassessing these associations using the large number of observations that such disaggregation would allow

Response: Another excellent point. We have disaggregated these correlations to use individual ROI sites rather than just the 7 ROIs. These ‘correlations’ now use logistic or linear mixed models instead of Pearson correlations, enabling us to account for between-subjects heterogeneity and within-subject effects (pg. 8, 9 & 11).

R2, Point 3: It is notable that vmPFC does not show greater activity for self > other, and if anything, trends in the other direction. This would seem to be at odds with a large number of fMRI findings indicating that this region is involved in self-relevant processing. Perhaps the present result is simply a false negative – we cannot know for certain. However, it would be interesting to hear consideration of this result in the discussion, and any potential explanation for the discrepancy between ECoG and fMRI in this regard.

Response: We have added a vmPFC discussion paragraph (pg. 17) that was originally cut from the manuscript, which mostly relates to schemas, but also relates to self/other differences:

“These early-onset vmPFC sites produced equivalent activations for all task conditions (Figs. 2k, 3c & 5a), consistent with rapid magnocellular gist processing, which likely cannot differentiate our alphanumeric stimuli (Fig. 1a). In contrast, late-onset vmPFC sites produced longer activations for other-mentalizing versus self-mentalizing (Figs. 5cd & 2f), aligning with other-mentalizing’s greater reliance on schematic feedback, especially in our trait judgement task. Nevertheless, self- and other-mentalizing recruited near-identical vmPFC sites (91% overlap; Fig. 5ab), suggesting common schematic underpinnings.”

Of note, our fMRI meta-analysis found other-selective and non-selective vmPFC areas, but no self-selective vmPFC areas (Lieberman et al., 2019; <https://doi.org/10.1016/j.neubiorev.2018.12.021>). This does not preclude vmPFC’s importance in self-referential processing; instead, the neurocognitive functions of vmPFC may be more heavily

taxed by social tasks. Moreover, the “vmPFC” described by many papers is actually in the location of our amPFC ROI, which may account for some discrepancies.

R2, Point 4: Page 2, line 8: “higher...” higher implies a comparison, but no other modality is mentioned in this sentence or the preceding or following one. Implicitly the comparison is to fMRI, but that comparison doesn't become explicit until later.

Response: Fixed the sentence to: “Several studies have investigated the fast spatiotemporal dynamics of mentalizing processing using source-space electroencephalography (EEG) and magnetoencephalography (MEG), neuroimaging modalities with millisecond temporal resolution but coarse spatial resolution.”

R2, Point 5: The bottom “latency” labels in figure 2 appear partially cut off.

Response: Thanks for pointing this out. This was an issue when converting to PDF format. We are now using a press-quality PDF converter and have also uploaded separate full-res 600dpi images.

R3 opening: This paper addresses the important question of the neural mechanisms for reasoning about ourselves and others (social neuroscience). The authors studied the spatio-temporal patterns of activation of the human brain during self-mentalizing, other-mentalizing, and mental calculations using a series of true-false questions. They use a localizer task while recording intracranial EEG recordings to localize contacts selective for mentalizing trials and analyze their response latency (both based on high-frequency gamma band power, HFB).

Results: First, the proportion of electrodes selectively responding to mentalizing trials increases from posterior to anterior (across visual, TPJ, ATL, PMC, PFC). Second, similar for the onset latency (Fig. 3), establishing a sequence of activations anterior to posterior. Third, all areas examined contained a mixture of self, other and both conditions (Fig 4), with differences between self and other in terms of latency of peak and duration of activation.

Positive: The conclusions are based on a large dataset of high quality. The results are novel and offer a comprehensive large-scale mapping of the location and response latency of contacts carrying self/other selective HFB activity. This task is much studied in social neuroscience and thus of high relevance.

Negatives: Design caveats in the task pose numerous caveats on the interpretation. Also puzzling are several bold claims are made that are not supported by this rather simple localizer task. There is no relationship between the neural activity assessed and behavior in a given trial, leaving the behavioral relevance of the activity observed unclear.

Response: We would like to give our thanks for your positive feedback and helpful negative feedback. We have made numerous changes in the analyses and manuscript text that address your important concerns.

R3, Point 1: Behavior and task. First, there is no analysis of behavior. How do we know the patients were successful at mentalizing about themselves or others (and doing mental math) ? What are the distribution of RTs, responses etc ? Second, the task seems not well designed to address the overall question. The three types of trials vary in several ways: reading words vs numbers, number of words, and RT. The fact that only the mentalizing conditions included words is a confound for the mentalizing vs cognitive contrasts – one could equally call this a “reading words vs numbers” contrast? Also at least in the example the number of words is higher for the ‘other mentalizing’ trials

Response: Thanks for these helpful and relevant comments. Analysis of behavior has been highlighted in the results (pg. 12). Distribution of RTs and response choices have been added in Fig. S4. To make clear that these are behavioral metrics, response times are now called RT_{Behav} , while response choices are called $\text{Choice}_{\text{Behav}}$

To highlight that we confirmed that subjects were paying attention, we have added these sentences to the Results (pg. 5) and Methods (pg. 20) sections: “For all analyses, trials were excluded if they met any of these criteria: 1) irrelevant button presses, 2) RT_{Behav} under 400 ms, or 3) no behavioral responses. Analyses of HFB timing metrics also excluded trials with RT_{Behav} over 5000 ms.”

Unfortunately, all other-mentalizing prompts are one word longer than self-mentalizing prompts (Table S4): “My neighbor is [trait]” versus “I am [trait]”. Similarly, all arithmetic prompts use numerals instead of words.

R3, Point 2: Relationship between RT and latency/power (Fig 4). First, these differences could be due to differences between the trials (task design, see above) rather than self vs other differences. RTs are longer in the other condition, so the effects seen in Fig 4c-e might be due to this. As controls, one would expect analysis where subsets of trials are selected where RT (and number of words, if that is possible) are matched, then examining whether the observed latency differences still exist. Using a mixed effect model to account for effects of RT does not solve this problem. Second, RT differences could be due to tuning to choices (true vs false). Are the proportion of true/false answers balanced between the self vs other groups? Third, the conclusion of 'stronger association with response times' with areas that respond later are not supported by Fig 4e - the associations seem similar in magnitude and this statement is based on a null result. Overall, Fig 4 was the least convincing part of the paper.

Response: All trial-by-trial ROI analyses of self/other differences (Fig. 5c-e, Table 1) controlled for RT_{Behav} by using RT_{Behav} as a random effect nested within Subject. This accounts for RT_{Behav} overall, and accounts for between-subject heterogeneity by including between-subject variance/covariance parameters across within-subject random effects. This is now better described in Results (pg.11) and Methods (pg. 20 & 26).

We have added a supplemental figure (Fig. S3) showing HFB latencies from behavior-matched trials across ROIs and mentalizing type. Mean RT_{Behav} was 2000 ± 5 ms within each ROI and mentalizing type. Trials were included only if subjects chose 'true' in response to task prompts. We did not perform formal statistical tests on these data, as only ~800 observations remained out of 23k+ observations in primary analyses. This figure is meant to be illustrative that the patterns observed in the main analyses remain stable when matched for behavior.

We also added analysis of RT_{Behav} across $\text{Choice}_{\text{Behav}}$ and mentalizing type, indicating that tuning to choices is secondary to self/other differences in RT_{Behav} (pg. 12):

“Faster RT_{Behav} was evoked by self-mentalizing ($M=2391 \pm 51$ ms) versus other-mentalizing ($M=2736 \pm 51$ ms; $F_{1,100}=22.6$, $p=6.65e-6$). Additionally, RT_{Behav} was faster for 'true' choices ($M=2457 \pm 45$ ms) versus 'false' choices ($M=2738 \pm 60$ ms; $F_{1,19}=4.82$, $p=.041$). However, $\text{Choice}_{\text{Behav}}$ effects did not differ between self-mentalizing ('true'=64%) and other-mentalizing ('true'=60%; interaction $F_{1,16}=0.86$, $p=.367$). In sum, mentalizing type was the strongest predictor of RT_{Behav} .”

We have disaggregated the correlations for “ROIs with later activations have stronger associations with behavioral responses”, which now use within-site RT_{Behav} effect sizes across individual ROI sites. This revealed that ROI sites with later activations have stronger RT_{Behav} effects in *temporal* HFB metrics (pg. 9):

“We found that sites with later onsets had stronger RT_{Behav} effects for peak latency ($t_{255}=11.079$, $p=1.51e-23$). In contrast, RT_{Behav} effects for peak power did not vary significantly by onset latency ($t_{255}=-0.101$, $p=.919$). Taken together, RT_{Behav} was better predicted by successive ROI sites in terms of activation latency (Fig. 4d) but not activation magnitude (Fig. 4e).”

R3, Point 3: Within vs across patient differences. How are differences across patients taken into account? i.e. presumably some patients had TPJ electrodes, and others had vmPFC electrodes. Could the latency differences be due to generally slower processing in some vs other patients? The mixed effect models do not appear to take into account between-subject variance.

Response: Pairwise ROI comparisons of HFB latencies only used subjects with simultaneous coverage in both ROIs. Pairwise LMEMs were changed to measure trial-by-trial ROI differences by nesting ROI within Trial within Subject. This constrains the effect of interest: the fixed effect for ROI differences. This is highlighted in Results (Pg. 7).

Accounting for between-subject variance is now better described in Methods (pg. 26-27): "All LMEMs included Subject as a nesting factor, which accounted for within-subject effects and between-subjects heterogeneity by including unconstrained between-subjects variance/covariance matrices across within-subject random effects. LMEMs that included Site as nesting factor likewise accounted for variance within and between sites."

LMEMs used for pairwise ROI comparisons and self/other comparisons in ROIs included RT_{Behav} as a random effect nested within Subject. This controls for RT_{Behav} and between-subject heterogeneity. This is now highlighted in Results (pg. 7 & 10) and Methods (pg. 26-27)

R3, Point 4: Claim of 'near identical populations' (made several times, including abstract). I cannot see how this is supported based on comparing gamma power. HFB is the sum of synaptic activity across a large number of neurons. This does not allow conclusion about which groups of neurons are activated. All that is supported is that gamma band power did not differ

Response: We have changed all references to 'near-identical neuronal populations' to 'near-identical cortical sites'.

R3, Point 5: Introduction/discussion. The introduction and discussion is simplistic and does not reflect the physiological basis of what is measured – it is assumed that fMRI and EEG/MEG/ECOG measure the same thing, but with different temporal and spatial resolution. There is certainly more than the 'limited temporal resolution of fMRI' that precludes fMRI from revealing a 'precise account of mentalizing and its accounts'. It would help the paper by properly framing the differences in what is measured here vs. what fMRI measures (metabolic activity).

Response: We have added this phrasing in Introduction (pg. 4): "...we recorded high-frequency broadband activity (HFB; 70-180 Hz), which reflects the rapid spiking of neuronal populations. In contrast, fMRI measures slow metabolic changes, although fMRI and HFB correspond in the anatomy and direction of measured effects (see Parvizi & Kastner, 2019)"

We have also added a caveat to one of the most scintillating points in the discussion regarding dmPFC other-selectivity (pg. 17): "Given that standard fMRI analysis does not distinguish activation intensity from activation duration, it appears that the latter has been mistaken for the former – though we cannot exclude the possibility of confounds related to differences between ECoG and fMRI"

R3, Point 6: Claim of a ‘common neurocognitive pathway’ that is ‘hierarchical’. I can't see how these claims are supported. What is shown is an impressive and very interesting set of latency and HFB power tuning differences across cortex. But how does this show a pathway and a hierarchy? Seeing latency differences does not make something ‘a pathway’ - this does not show that these areas communicate with each other (pathway) nor that they form a hierarchy. Similar for the statement of ‘ascending through’ and similar in the discussion – I cannot see how the data support that. Trial-by-trial analysis of latency differences across simultaneously recorded contacts would be needed to support these claims.

Response: Many thanks for the kind words on the results. We have changed existing analyses and included additional analyses in response to your comments. We think these changes make for a much more convincing set of results – thanks!

As mentioned above: pairwise ROI comparisons of HFB latencies only used subjects with simultaneous coverage in both ROIs. Pairwise LMEMs were changed to provide within-trial estimates of ROI latencies by nesting ROI within Trial within Subject (Pg. 7).

Critically, we also added supplementary analysis of within-trial onset differences across ROI groups: visual cortex, temporoparietal DMN, and mPFC (Fig. S2). This analysis used a purely random effects model with ROIgroup nested within Trial within Subject. This avoids any between-trial aggregation involved when including fixed effects. This was performed twice, first using subjects with simultaneous coverage in 2+ ROI groups, and then using subjects with simultaneous coverage in all ROI groups. A robust within-trial trend was found in both analyses, revealing consistent within-trial propagation of activation onsets: Visual→Temporoparietal DMN→mPFC.

From Fig. S2 caption: “When using subjects with coverage in 2+ ROI groups, we found a robust within-trial trend of onset latency differences across ROI groups ($F_{1,1538}=3408$, $p<2.23e-308$; exact p -value below machine precision). For subjects with coverage in all ROI groups, the trend contrast was also robust ($F_{1,862}=2642$, $p=1.04e-264$). These results reveal a consistent within-trial sequence of mentalizing activations, which began in visual cortex, then spread to temporoparietal DMN regions, and finally to medial prefrontal regions.”

Lastly, we added RT-locked analysis of activation offsets (pg. 7): “We also performed post-hoc analysis of offset latencies relative to RT_{Behav} (Fig. 4c), revealing that mPFC activations more closely preceded RT_{Behav} than other ROIs combined ($b_{20204}=138$ ms, $SE=31$, $p=4.28e-6$).”

Taken together, we believe that our results portray a neurocognitive pathway. Trial-by-trial ROI onset differences show the initial feedforward propagation of activations across the pathway. This pathway follows a gradient of functional specialization across ROI sites: later onset latencies robustly predicted greater mentalizing-specificity (Fig. 4b), greater temporal associations with behavioral responses (pg. 8-9 & Fig. 4d), and greater self/other temporal differentiation (pg. 11 & Fig. 5cd).

REVIEWER COMMENTS

Reviewer #1 (Remarks to the Author):

The authors have worked hard to address all of my and the other reviewers' critiques. I'm especially impressed at their new analyses and am happy with their clarifications of the points/questions raised on the last round of review. I would support publication at this point.

Reviewer #2 (Remarks to the Author):

The authors have satisfactorily addressed all of the concerns I raised in my initial review. I believe this paper is now poised to make a valuable contribution to our understanding of mentalizing.

Reviewer #3 (Remarks to the Author):

This is a revision of a paper I reviewed before. While the authors addressed some of the issues, the most major issues were essentially ignored and glanced over. At the very least the authors need to convincingly argue why a given point was not addressed (it seems like an odd strategy which I see the authors also used to reply to the other reviewers to reply with analysis not addressing the question).

At present I do not think the major claim of 'our results reveal a common neurocognitive pathway for self-and other mentalizing that follows...' is supported because there is no evidence showing that the patients do any self or other mentalizing (nor that they do mental calculation) and there are no controls to substantiate the major claim that self-and other mentalizing recruit a common substrate. As such this is a fine and very well done localizer study, but its specificity and relevance to mentalizing remains unclear. As such I do not believe this raises to the conceptual advance needed for this journal.

1.Behavior. My question was: "How do we know the patients were successful at mentalizing about themselves or others (and doing mental math) ?" I realize there is RT analysis, but the most critical aspect of a behavioral analysis is in analyzing whether patients are performing the task they are told to perform (accuracy). Excluding trials with too long or too short RTs does not show that patients paid attention to task or performed the task they were asked to do. As things stand a patient could give random responses without understanding the task and they would still be considered to be 'mentalizing'.

2.Task. The differences observed could be due to differences between the trials (task design, see above) rather than self vs other differences. A critical control that is missing is involving reading/answering questions that do not involve mentalizing. As such a 'mentalizing response' here could be a non-specific visual response to words. Same for self. vs other since the stimuli were different.

3. Claim of a 'common neurocognitive pathway for mentalizing'. This is a localizer task that shows where and when HFB power changes when reading questions about self vs other. This does not establish a pathway so I urge the authors to re-word the title and text to reflect what this data shows.

Dear reviewers,

We are grateful for the continued interest in our manuscript, especially the enthusiastic support for publication now given by two of three reviewers. We have made every possible effort to ameliorate all remaining reviewer concerns.

Our main changes were as follows:

- 1) Providing strong evidence that subjects attentively performed the mentalizing task by analyzing behavioral response choices ($\text{Choice}_{\text{Behav}}$). We show that $\text{Choice}_{\text{Behav}}$ had differential self/other biases towards positive or negative affective traits, indicating that subjects discerned the target and trait of mentalizing prompts in ways consistent with actual mentalization.
- 2) Providing additional evidence for task engagement by showing very high accuracy in arithmetic trials (median = 92.5%).
- 3) Accounting for stimulus visual dissimilarity (VD) by analyzing task prompts with a computer vision model. After controlling for VD, mentalizing-related effects (self/other differences, RT_{Behav} , and $\text{Choice}_{\text{Behav}}$) were still significant in DMN ROIs, indicating that such effects were not explained by low-level differences across stimuli.
- 4) Changing the term “neurocognitive pathway” to “neurocognitive sequence” in response to Reviewer Three’s persistent concerns.

We also made minor changes in the manuscript for typos, clarity, and word count.

Point-by-point responses to reviewer concerns are found below, with page breaks after each reviewer.

Sincerely,

Kevin Tan and co-authors

R1 opening: “The authors have worked hard to address all of my and the other reviewers' critiques. I'm especially impressed at their new analyses and am happy with their clarifications of the points/questions raised on the last round of review. I would support publication at this point.”

Response: We would like to give many thanks for your support regarding publication, along with for your kind words and helpful suggestions throughout the review process!

R2, Opening: “The authors have satisfactorily addressed all of the concerns I raised in my initial review. I believe this paper is now poised to make a valuable contribution to our understanding of mentalizing”

Response: Thank you very much for your support regarding publication, as well as your helpful comments and suggestions throughout the review process!

R3, Opening: “This is a revision of a paper I reviewed before. While the authors addressed some of the issues, the most major issues were essentially ignored and glanced over. At the very least the authors need to convincingly argue why a given point was not addressed (it seems like an odd strategy which I see the authors also used to reply to the other reviewers to reply with analysis not addressing the question).

At present I do not think the major claim of ‘our results reveal a common neurocognitive pathway for self-and other mentalizing that follows...’ is supported because there is no evidence showing that the patients do any self or other mentalizing (nor that they do mental calculation) and there are no controls to substantiate the major claim that self-and other mentalizing recruit a common substrate. As such this is a fine and very well done localizer study, but its specificity and relevance to mentalizing remains unclear. As such I do not believe this raises to the conceptual advance needed for this journal.”

Response: We are sorry you feel that we ignored your points in the previous revision. We previously tried to thoroughly address your points by spending months on additional analyses and re-analysis, and by writing a lengthy response letter. Considering the specific issues raised in this round of reviews, we have conducted several additional analyses that we believe address each of your concerns.

R3, Point 1: “Behavior. My question was: “How do we know the patients were successful at mentalizing about themselves or others (and doing mental math)?” I realize there is RT analysis, but the most critical aspect of a behavioral analysis is in analyzing whether patients are performing the task they are told to perform (accuracy). Excluding trials with too long or too short RTs does not show that patients paid attention to task or performed the task they were asked to do. As things stand a patient could give random responses without understanding the task and they would still be considered to be ‘mentalizing’.”

Response: Mentalizing studies using trait judgment tasks predominantly do not consider accuracy, given that 1) trait judgments are subjective, and 2) mentalizing targets are often unique to each subject. For a recent review of social cognition neuroimaging studies ($n=188$ studies), including trait judgment studies ($n=19$ studies), see Schurz et al., 2021. Many of these studies do not consider accuracy or even RTs.

In lieu of accuracy metrics for mentalizing, we examined $\text{Choice}_{\text{Behav}}$ for self/other biases towards positive or negative affective traits (pg. 12-13 & Fig. S4c). If subjects were inattentive to the mentalizing prompts, we would expect random responses or responses that were insensitive to the target of the prompt (i.e., self/other). Past social cognition studies reliably find a positivity asymmetry in attributions for self and others, such that we attribute more positive characteristics to the self than to others. To examine this, we divided our stimuli into positive and negative traits. As now reported in the manuscript, we indeed saw the classic pattern of greater positivity bias for the self than for the other. This is strong evidence that subjects were following instructions and performing the task as expected (Results, pg. 12-13):

“To confirm that subjects performed mentalizing for task prompts, we analyzed $\text{Choice}_{\text{Behav}}$ for self/other biases towards positive or negative affective traits (Fig. S4c) using logistic mixed-

effects classification (nested within Trait within Subject). Overall, we found a greater probability of ‘true’ choices for positive versus negative traits (odds ratio₁₄₇₁=2.53±.071, $p=1.04e-39$). This bias was stronger during self-mentalizing versus other-mentalizing (odds ratio₁₄₇₁=1.38±.066, $p=4.56e-7$). These results indicate that subjects engaged the mentalistic content in task prompts with a self-positivity bias, which is a canonical feature of mentalizing³⁷⁻³⁹. Moreover, high accuracy in arithmetic trials (median=92.5%; Fig. S4b) suggests effortful attention to task prompts.”

Additionally, we examined accuracy for arithmetic trials, revealing very high accuracy across subjects (median=92.5%; Fig. S4b). It seems implausible that subjects would selectively engage in accurate responding for arithmetic trials, but not mentalizing trials, given that our mentalizing prompts are very simple and clear.

R3, Point 2: “Task. The differences observed could be due to differences between the trials (task design, see above) rather than self vs other differences. A critical control that is missing is involving reading/answering questions that do not involve mentalizing. As such a ‘mentalizing response’ here could be a non-specific visual response to words. Same for self. vs other since the stimuli were different.”

Response: To comprehensively control for visual dissimilarity across stimuli, we used a computer vision model to extract two dimensions of visual dissimilarity (VD1 and VD2). VD1 represented prompt length, while VD2 represented other visual features (Fig. S1). VD1 and VD2 were used as ‘nuisance’ covariates in ROI analyses (see Supplementary Methods).

If self/other differences were solely due to prompt length or other visual differences, one would expect self/other HFB differences to be non-dissociable from VDs, with the greatest VD effects on HFB latencies in mPFC where self/other differences were strongest. After controlling for VD, previously-observed self/other differences were still significant in DMN ROIs. Moreover, supplementary analysis of VD encoding found no VD effects in mPFC ROIs (Supplementary Table S1). In other words, all previously-observed self/other differences still held after controlling for VDs, except for visual cortex, which is not part of the mentalizing network.

We added the following paragraph in Results (pg. 12): “To distinguish mentalizing-related neuronal effects (mentalizing type, RT_{Behav} , and $\text{Choice}_{\text{Behav}}$) from stimulus VD (e.g., prompt length), we used computer vision^{36,37} (see Supplementary Methods). After controlling for VD, mentalizing-related effects were still significant in all DMN ROIs (Figs. 4de, 5c-e & Table 1). However, in visual cortex, marked self/other differences in peak power became nonsignificant (Fig. 5e). In sum, mentalizing-related effects in DMN ROIs were not explained by prompt length and other visual features.” VD results are further elaborated in Supplementary Materials (pg. 3).

If mentalizing-specificity was solely due to differences between sentences and numerical equations, one would expect mentalizing-specific sites to be clustered in the ‘reading network’ (see ECoG reading studies: Llorens et al., 2011; Wu et al., 2011; Miller et al., 2011; Long et al., 2020 & fMRI meta-analysis: <https://neurosynth.org/analyses/terms/reading/>).

Yet, we found that mentalizing-specific sites (Fig. 3c) were not clustered in the 'reading network', but instead in the 'mentalizing network' of DMN regions found by countless fMRI studies (see Schurz et al., 2021 for a recent review). Given the remarkable consistency between our mentalizing-specificity results and fMRI mentalizing studies, we find little reason to believe that our results are attributable to reading instead of mentalizing.

The following sentences were added to the limitations section (pg. 20): "Another task-related confound was greater prompt length for other-mentalizing (e.g., "My neighbor is...") versus self-mentalizing (e.g., "I am..."; Table S4), which we controlled for using computer vision^{36,37} (see Supplementary Methods). Relatedly, mentalizing-specificity (Fig. 3cd) could arise from differences between sentences and arithmetic equations. However, mentalizing-specific sites were not concentrated in reading-related regions¹¹⁸⁻¹²¹, but rather in the 'mentalizing network' reported by countless fMRI studies¹²."

R3, Point 3: "Claim of a 'common neurocognitive pathway for mentalizing'. This is a localizer task that shows where and when HFB power changes when reading questions about self vs other. This does not establish a pathway so I urge the authors to re-word the title and text to reflect what this data shows."

Response: In your initial review, you claimed that "trial-by-trial analysis of latency differences across simultaneously recorded contacts would be needed to support these claims." In response, we added supplementary analysis showing a robust trial-by-trial sequence in the propagation of activation onsets: visual cortex → temporoparietal DMN → mPFC (Fig. S3). This occurred in subjects with simultaneous coverage in all three ROI groups. Additionally, this analysis even controlled for subject- and trial-related heterogeneity.

Moreover, in the previous revision, we modified the pairwise ROI comparisons (Fig. 3f) to measure trial-by-trial latency differences, controlled for subject- and trial-related heterogeneity. Critically, pairwise comparisons were always restricted to subjects with simultaneous coverage in both ROIs.

Nevertheless, we have changed "neurocognitive pathway" to "neurocognitive sequence" in the title and results section. We keep some hedged references of a "putative neurocognitive pathway" in two paragraphs of the discussion, also stating that "further research involving connectivity analyses are needed for more conclusive claims"

REVIEWER COMMENTS

Reviewer #2 (Remarks to the Author):

The authors have made a number of revisions to address lingering issues raised by reviewer 3. Their analysis of positive vs. negative trait endorsement for self vs. other provides useful additional evidence that participants were indeed engaging with this task as intended. Their examination of visual similarity between stimuli helps to rule out potential low-level confound that could account for their results. The authors have also tempered their language regarding the use of the term 'pathway' to address R3's concern about this point. To me it seems as though the authors have made useful changes that have addressed these remaining concerns.

Reviewer #3 (Remarks to the Author):

The authors fully addressed my remaining concerns and I have no further requests. Congratulations on an excellent paper.

I would like to commend the authors for the thoughtful and comprehensive new analysis offered to my critique. The behavioral analysis of a self vs. other bias in 'true' judgments is interesting and a very valuable addition, as is the computer-vision feature based control for mentalizing specific activity (which, I note, did eliminate visual cortex).

As a note (no change required due to this), the authors note that most mentalizing studies in social cognitive neuroscience do not consider behavior. While true this does not mean it is the right thing to do, especially given the poor replicability track record of this field. Indeed, from painful experience, I have learned that subjects can and often do not do the task instructed, so the kind of behavioral accuracy support the authors added is absolutely critical to make the kinds of statements that are advanced in this paper. This fact could be highlighted as a unique strength of this paper in discussion.

Dear reviewers,

We are thankful for the continued interest and conditional acceptance of our manuscript.

We made slight changes to the manuscript given editorial requests, typos, and clarity. Point-by-point responses to the reviewers are found below.

Sincerely,

Kevin Tan and co-authors

Reviewer #2: “The authors have made a number of revisions to address lingering issues raised by reviewer 3. Their analysis of positive vs. negative trait endorsement for self vs. other provides useful additional evidence that participants were indeed engaging with this task as intended. Their examination of visual similarity between stimuli helps to rule out potential low-level confound that could account for their results. The authors have also tempered their language regarding the use of the term 'pathway' to address R3's concern about this point. To me it seems as though the authors have made useful changes that have addressed these remaining concerns.”

Response: We would like to give many thanks for your support regarding publication, along with for your kind words and helpful suggestions throughout the review process!

Reviewer #3: “The authors fully addressed my remaining concerns and I have no further requests. Congratulations on an excellent paper.

I would like to commend the authors for the thoughtful and comprehensive new analysis offered to my critique. The behavioral analysis of a self vs. other bias in 'true' judgments is interesting and a very valuable addition, as is the computer-vision feature based control for mentalizing specific activity (which, I note, did eliminate visual cortex).

As a note (no change required due to this), the authors note that most mentalizing studies in social cognitive neuroscience do not consider behavior. While true this does not mean it is the right thing to do, especially given the poor replicability track record of this field. Indeed, from painful experience, I have learned that subjects can and often do not do the task instructed, so the kind of behavioral accuracy support the authors added is absolutely critical to make the kinds of statements that are advanced in this paper. This fact could be highlighted as a unique strength of this paper in discussion.”

Response: We are thankful for your kind words and support regarding publication! We are also thankful for your rigorous critiques and suggestions throughout the review process, which have resulted in a stronger manuscript.